- UNCERTAINTIES IN THE EDGAR EMISSION INVENTORY OF GREENHOUSE GASES
- Efisio Solazzo<sup>1</sup>, Monica Crippa<sup>1</sup>, Diego Guizzardi<sup>1</sup>, Marilena Muntean<sup>1</sup>, Margarita Choulga<sup>2</sup>, Greet
   Janssens-Maenhout<sup>1</sup>
- <sup>4</sup> <sup>1</sup> European Commission, Joint Research Centre, JRC, Ispra (VA), 21027, Italy
- <u>2 Research Department</u>, European Centre for Medium Range Weather Forecasting, <u>ECMWF</u>, Reading,
- <u>RG2 9AX, United Kingdom</u>
- Abstract

7

9 The Emissions Database for Global Atmospheric Research (EDGAR) estimates the human-induced 10 emission rates on Earth, EDGAR collaborates with atmospheric modelling activities and aids policy in 11 the design of mitigation strategies and in evaluating their effectiveness. In these applications, the 12 uncertainty estimate is an essential component, as it quantifies the accuracy and qualifies the level of

13 confidence in the emission.

This study complements the EDGAR's emissions inventory with estimation of the structural uncertainty stemming from its base components (activity data statistics (AD) and emission factors (EF<sub>4</sub>), by *i*) associating uncertainty to each AD and EF characterizing the emissions of the three main greenhouse gases (GHGs), namely carbon dioxide (CO<sub>2</sub>), methane (CH<sub>4</sub>) and nitrous oxide (N<sub>2</sub>O<sub>2</sub>; *ii*) combining

them, and *iii*) making assumptions for the cross-country uncertainty aggregation of source categories.

19 It was deemed a natural choice to obtain the uncertainties in EFs and AD from the Intergovernmental

Panel on Climate Change (IPCC) guidelines issued in 2006 (with a few exceptions), since the EF and AD sources and methodological aspects used by EDGAR have been built over the years based on the IPCC recommendations, which assured consistency in time and comparability across countries. While on one side the homogeneity of the method is one of the key strengths of EDGAR, on the other side it facilitates the propagation of uncertainties when similar emission sources are aggregated. For this reason, this study aims primarily at addressing the aggregation of uncertainties sectorial emissions across GHGs and countries.

<u>Globally</u>, we find that the anthropogenic emissions covered by EDGAR of the combined three main 28 GHGs for the year 2015 are accurate within an interval of -15% to +20% (defining the 95% confidence 29 of a log-normal distribution). The most uncertain emissions are those related to N<sub>2</sub>O from <u>waste and</u> 30 agriculture, while CO<sub>2</sub> emissions, although responsible for 74% of <u>the total GHG emissions</u>, <u>account</u> 31 for approximately 11% of global uncertainty share. Sensitivity to methodological choices is also 32 discussed.

### 33 <u>1</u>INTRODUCTION

According to the latest release of the Emissions Database of Global Atmospheric Research (EDGAR

version 5, *Crippa et al.*, 2019; *Crippa et al.*, 2020a), in the year 2015 the global greenhouse gas (GHG)

- emissions of <u>carbon dioxide</u> ( $CO_2$ ), <u>methane</u> ( $CH_4$ ) and <u>nitrous oxide</u> ( $N_2O$ ) due to anthropogenic
- activities <u>summed up to 48.1 Gt CO<sub>2</sub>eq</u> (CO<sub>2</sub> equivalent emissions (CO<sub>2</sub>eq) are computed using the
   Global Warming Potential values from the Fourth Assessment Report (AR4) of the Intergovernmental
- Global Warming Potential values from the Fourth Assessment Report (AR4) of the Intergovernmental
   Panel on Climate Change (IPCC)). In the same year, the share of global CO<sub>2</sub>eq. from non-CO<sub>2</sub> GHG
- emissions (i.e.  $CH_4$  and  $N_2O$ ) was approximately <u>a quarter</u>. Measures put in place to attenuate

temperature rise and to mitigate climate dynamics long-term changes, have contributed to uphold the

| -{ | ha eliminato: <sup>2</sup>   |
|----|------------------------------|
| 1  | ha eliminato: Shinfield Park |
| 4  | ha eliminato: UK             |

| ha eliminato: collaborating     |
|---------------------------------|
| ha eliminato: as well as aiding |

| - | ha eliminato: ))   |
|---|--------------------|
|   |                    |
| - | ha eliminato: GHG) |
| 1 | ha eliminato: ,    |
| Y | ha eliminato: ;    |
| Ν | ha eliminato:      |

| 1  | ha eliminato: On global average                                        |
|----|------------------------------------------------------------------------|
| X  | ha eliminato: accounts for and                                         |
|    | ha eliminato: ,                                                        |
| /( | ha eliminato: totaled                                                  |
| // | ha eliminato: <sup>1</sup> .                                           |
| // | ha eliminato: CO <sub>2</sub> -equivalent (                            |
| // | ha eliminato: ) of                                                     |
| // | ha eliminato: of 1/4. Significant efforts expanded to promote measures |
| Å  | ha formattato: Evidenziato                                             |
| // | ha eliminato: rising                                                   |
| Å  | ha formattato: Evidenziato                                             |
| Å  | ha formattato: Evidenziato                                             |
| 1  | ha eliminato: long-term change to                                      |
| 1  | ha formattato: Evidenziato                                             |
| -  | ha formattato: Evidenziato                                             |

role of CH4 and N2O. Their high warming potential compared to CO2 and relatively shorter life-time 63 (on average  $CH_4$  persists in the atmosphere for approximately a decade,  $N_2O$  for over a century and 64 65 CO<sub>2</sub> for more than 1000 years (NCR, 2010; Ciais et al., 2013)) allow to act on shifting from energy-66 related CO<sub>2</sub> to other, more rapidly responsive, emission sources (Janssens-Maenhout et al., 2019; United Nations Environment Programme, 2019). At the same time, while for fossil fuel CO<sub>2</sub> emissions 67 the uncertainty is relatively small and, overall, well defined, for <u>CH<sub>4</sub> and N<sub>2</sub>O</u> the emission estimates 68 are significantly more uncertain. In turn, emission reduction measures issued by national plans highly 69 70 depend on the degree of uncertainty of sectors that are supposed to contribute to reach the designed 71 reduction <u>target</u>. As depicted in the example by *Olivier (1998)* a sector contributing by 10% to the 72 national reduction target may contribute to 5% or 15% if that sector's emission factor is ±50% uncertain. 73 EDGAR aims to consolidate its position in supporting research and new data/approach implementation 74 in operational modelling, as well as becoming an independent tool supporting policy makers in

monitoring and mitigation <u>strategies</u>. Therefore, a reliable quantification of the uncertainties <u>should</u>
have the same degree of importance as the consistency and comparability of the emissions. This study
<u>evolves</u> in this direction, by adding the uncertainty dimension to the EDGAR database, thus enhancing
its value with much needed information on reliability, and promote comparability with other datasets.
<u>Uncertainty reports are relevant</u>, among other applications, for:

- scientific purposes, e.g. assessing robustness of long-term emission trends, or provide a-priori
   state for comparison with independent top-down estimates (*Bergamaschi et al.*, 2018), or aid
   in network design (*Super et al.*, 2020);
  - inter-comparison studies (Choulga et al., 2020; Petrescu et al., 2020);
  - assessing the feasible potential of mitigation strategies (e.g. Van Dingenen et al., 2017).

This study adds the uncertainty component to the EDGAR data by devising methods to propagate the 85 86 uncertainty introduced by activity data (AD) and emission factors (EFs) to any combination/aggregation 87 of sources, countries, and GHGs. Methods, aggregation strategies and dependencies are presented and 88 investigated. Analyses are conducted for the emission year 2015 for CO<sub>2</sub>, CH<sub>4</sub> and N<sub>2</sub>O. Sensitivity to 89 methodological choices is also discussed. The methodology presented here has been already applied to 90 EDGAR and discussed in the scientific literature in comparison to other methods (Choulga et al., 2020), 91 to other inventories (Petrescu et al., 2020), to assess the uncertainty of the EDGAR-FOOD inventory (Crippa et al., <u>2021</u>), applied to specific sectors (Muntean et al., <u>2021</u>), trend analysis of global GHG 92 emissions and to communicate with the policy makers and the public (Crippa et al., 2019, 2020c). 93

2 METHODOLOGY

83

84

EDGAR is a 'bottom-up' model for estimating emissions, <u>relying</u> on a large spectrum of <u>AD</u> covering human activities with a high degree of detail. AD are combined with <u>EFs</u> to yield the emission, per source, and country. For example, for combustion sources AD consist of fossil fuel consumption while the EF is the amount of emission produced per unit of activity. In this case the emission is typically obtained simply by multiplying AD by EF, while other sources (e.g. waste) require more sophisticated models.

AD are primarily retrieved from international statistics, complemented, when necessary, with information (e.g. trends) from other sources, such as scientific literature and national data. The quality, consistency, and comparability of AD through time and space are the essential features defining the quality, of an emission database.

| 1                | ha eliminato: non-CO <sub>2</sub> gases, such as                                                                                                                                                                                                                                                                                                                                                                                                                                                                                                                                                                                                                                                                                                                                                                                                                                                                                                                                                                                                                                                                                                                                                                                                                                                                                                                                                                                                                                                                                                                                                                                                                                                                                                                                                                                                                                                                                                                                                                                                                                                                            |           |
|------------------|-----------------------------------------------------------------------------------------------------------------------------------------------------------------------------------------------------------------------------------------------------------------------------------------------------------------------------------------------------------------------------------------------------------------------------------------------------------------------------------------------------------------------------------------------------------------------------------------------------------------------------------------------------------------------------------------------------------------------------------------------------------------------------------------------------------------------------------------------------------------------------------------------------------------------------------------------------------------------------------------------------------------------------------------------------------------------------------------------------------------------------------------------------------------------------------------------------------------------------------------------------------------------------------------------------------------------------------------------------------------------------------------------------------------------------------------------------------------------------------------------------------------------------------------------------------------------------------------------------------------------------------------------------------------------------------------------------------------------------------------------------------------------------------------------------------------------------------------------------------------------------------------------------------------------------------------------------------------------------------------------------------------------------------------------------------------------------------------------------------------------------|-----------|
|                  | <b>ha eliminato:</b> (25 for CH <sub>4</sub> and 298 for N <sub>2</sub> O, over a time horizon of 100 years ( <i>IPCC</i> , 2007))                                                                                                                                                                                                                                                                                                                                                                                                                                                                                                                                                                                                                                                                                                                                                                                                                                                                                                                                                                                                                                                                                                                                                                                                                                                                                                                                                                                                                                                                                                                                                                                                                                                                                                                                                                                                                                                                                                                                                                                          |           |
|                  | ha formattato: Evidenziato                                                                                                                                                                                                                                                                                                                                                                                                                                                                                                                                                                                                                                                                                                                                                                                                                                                                                                                                                                                                                                                                                                                                                                                                                                                                                                                                                                                                                                                                                                                                                                                                                                                                                                                                                                                                                                                                                                                                                                                                                                                                                                  |           |
|                  | ha formattato: Evidenziato                                                                                                                                                                                                                                                                                                                                                                                                                                                                                                                                                                                                                                                                                                                                                                                                                                                                                                                                                                                                                                                                                                                                                                                                                                                                                                                                                                                                                                                                                                                                                                                                                                                                                                                                                                                                                                                                                                                                                                                                                                                                                                  |           |
|                  | ha eliminato: even                                                                                                                                                                                                                                                                                                                                                                                                                                                                                                                                                                                                                                                                                                                                                                                                                                                                                                                                                                                                                                                                                                                                                                                                                                                                                                                                                                                                                                                                                                                                                                                                                                                                                                                                                                                                                                                                                                                                                                                                                                                                                                          |           |
| $\left( \right)$ | ha formattato: Evidenziato                                                                                                                                                                                                                                                                                                                                                                                                                                                                                                                                                                                                                                                                                                                                                                                                                                                                                                                                                                                                                                                                                                                                                                                                                                                                                                                                                                                                                                                                                                                                                                                                                                                                                                                                                                                                                                                                                                                                                                                                                                                                                                  |           |
| Q                | ha eliminato: design mitigation strategies focusing                                                                                                                                                                                                                                                                                                                                                                                                                                                                                                                                                                                                                                                                                                                                                                                                                                                                                                                                                                                                                                                                                                                                                                                                                                                                                                                                                                                                                                                                                                                                                                                                                                                                                                                                                                                                                                                                                                                                                                                                                                                                         |           |
|                  | ha formattato: Evidenziato                                                                                                                                                                                                                                                                                                                                                                                                                                                                                                                                                                                                                                                                                                                                                                                                                                                                                                                                                                                                                                                                                                                                                                                                                                                                                                                                                                                                                                                                                                                                                                                                                                                                                                                                                                                                                                                                                                                                                                                                                                                                                                  |           |
|                  | ha eliminato: emission control measures                                                                                                                                                                                                                                                                                                                                                                                                                                                                                                                                                                                                                                                                                                                                                                                                                                                                                                                                                                                                                                                                                                                                                                                                                                                                                                                                                                                                                                                                                                                                                                                                                                                                                                                                                                                                                                                                                                                                                                                                                                                                                     |           |
|                  | ha formattato: Evidenziato                                                                                                                                                                                                                                                                                                                                                                                                                                                                                                                                                                                                                                                                                                                                                                                                                                                                                                                                                                                                                                                                                                                                                                                                                                                                                                                                                                                                                                                                                                                                                                                                                                                                                                                                                                                                                                                                                                                                                                                                                                                                                                  |           |
|                  | ha eliminato: less controversial and                                                                                                                                                                                                                                                                                                                                                                                                                                                                                                                                                                                                                                                                                                                                                                                                                                                                                                                                                                                                                                                                                                                                                                                                                                                                                                                                                                                                                                                                                                                                                                                                                                                                                                                                                                                                                                                                                                                                                                                                                                                                                        |           |
|                  | ha formattato                                                                                                                                                                                                                                                                                                                                                                                                                                                                                                                                                                                                                                                                                                                                                                                                                                                                                                                                                                                                                                                                                                                                                                                                                                                                                                                                                                                                                                                                                                                                                                                                                                                                                                                                                                                                                                                                                                                                                                                                                                                                                                               | [         |
|                  | ha eliminato: the other gases                                                                                                                                                                                                                                                                                                                                                                                                                                                                                                                                                                                                                                                                                                                                                                                                                                                                                                                                                                                                                                                                                                                                                                                                                                                                                                                                                                                                                                                                                                                                                                                                                                                                                                                                                                                                                                                                                                                                                                                                                                                                                               |           |
|                  | ha formattato: Evidenziato                                                                                                                                                                                                                                                                                                                                                                                                                                                                                                                                                                                                                                                                                                                                                                                                                                                                                                                                                                                                                                                                                                                                                                                                                                                                                                                                                                                                                                                                                                                                                                                                                                                                                                                                                                                                                                                                                                                                                                                                                                                                                                  |           |
|                  | ha eliminato: factor                                                                                                                                                                                                                                                                                                                                                                                                                                                                                                                                                                                                                                                                                                                                                                                                                                                                                                                                                                                                                                                                                                                                                                                                                                                                                                                                                                                                                                                                                                                                                                                                                                                                                                                                                                                                                                                                                                                                                                                                                                                                                                        |           |
|                  | ha formattato                                                                                                                                                                                                                                                                                                                                                                                                                                                                                                                                                                                                                                                                                                                                                                                                                                                                                                                                                                                                                                                                                                                                                                                                                                                                                                                                                                                                                                                                                                                                                                                                                                                                                                                                                                                                                                                                                                                                                                                                                                                                                                               | [         |
|                  | ha eliminato: targets                                                                                                                                                                                                                                                                                                                                                                                                                                                                                                                                                                                                                                                                                                                                                                                                                                                                                                                                                                                                                                                                                                                                                                                                                                                                                                                                                                                                                                                                                                                                                                                                                                                                                                                                                                                                                                                                                                                                                                                                                                                                                                       |           |
|                  | support to scientificupporting research and application<br>toew data/approach implementation in operational<br>modelling, as well as becoming an independent tool in<br>support ofupporting policy makers in monitoring and<br>mitigation policiestrategies. Therefore, a reliable                                                                                                                                                                                                                                                                                                                                                                                                                                                                                                                                                                                                                                                                                                                                                                                                                                                                                                                                                                                                                                                                                                                                                                                                                                                                                                                                                                                                                                                                                                                                                                                                                                                                                                                                                                                                                                          | 1 111     |
|                  | quantification of the uncertainties assumeshould have th                                                                                                                                                                                                                                                                                                                                                                                                                                                                                                                                                                                                                                                                                                                                                                                                                                                                                                                                                                                                                                                                                                                                                                                                                                                                                                                                                                                                                                                                                                                                                                                                                                                                                                                                                                                                                                                                                                                                                                                                                                                                    | (         |
|                  | quantification of the uncertainties assumeshould have th<br>ha formattato: Evidenziato                                                                                                                                                                                                                                                                                                                                                                                                                                                                                                                                                                                                                                                                                                                                                                                                                                                                                                                                                                                                                                                                                                                                                                                                                                                                                                                                                                                                                                                                                                                                                                                                                                                                                                                                                                                                                                                                                                                                                                                                                                      | •         |
|                  | quantification of the uncertainties assumeshould have th<br>ha formattato: Evidenziato<br>ha eliminato: /assessment/impact                                                                                                                                                                                                                                                                                                                                                                                                                                                                                                                                                                                                                                                                                                                                                                                                                                                                                                                                                                                                                                                                                                                                                                                                                                                                                                                                                                                                                                                                                                                                                                                                                                                                                                                                                                                                                                                                                                                                                                                                  | •         |
|                  | <pre>quantification of the uncertainties assumeshould have th ha formattato: Evidenziato ha eliminato: /assessment/impact ha formattato: Evidenziato</pre>                                                                                                                                                                                                                                                                                                                                                                                                                                                                                                                                                                                                                                                                                                                                                                                                                                                                                                                                                                                                                                                                                                                                                                                                                                                                                                                                                                                                                                                                                                                                                                                                                                                                                                                                                                                                                                                                                                                                                                  | <b>(</b>  |
|                  | quantification of the uncertainties assumeshould have th<br>ha formattato: Evidenziato<br>ha eliminato: /assessment/impact<br>ha formattato: Evidenziato<br>ha eliminato: as for example                                                                                                                                                                                                                                                                                                                                                                                                                                                                                                                                                                                                                                                                                                                                                                                                                                                                                                                                                                                                                                                                                                                                                                                                                                                                                                                                                                                                                                                                                                                                                                                                                                                                                                                                                                                                                                                                                                                                    | •         |
|                  | quantification of the uncertainties assumeshould have th<br>ha formattato: Evidenziato<br>ha eliminato: /assessment/impact<br>ha formattato: Evidenziato<br>ha eliminato: as for example<br>ha formattato                                                                                                                                                                                                                                                                                                                                                                                                                                                                                                                                                                                                                                                                                                                                                                                                                                                                                                                                                                                                                                                                                                                                                                                                                                                                                                                                                                                                                                                                                                                                                                                                                                                                                                                                                                                                                                                                                                                   | (<br>(    |
|                  | quantification of the uncertainties assumeshould have the ha formattato: Evidenziato ha eliminato: /assessment/impact ha formattato: Evidenziato ha eliminato: as for example ha formattato ha eliminato: to and a-posteriori ha eliminato: to an | (         |
|                  | <pre>quantification of the uncertainties assumeshould have th ha formattato: Evidenziato ha eliminato: /assessment/impact ha formattato: Evidenziato ha eliminato: as for example ha formattato ha eliminato: to and a-posteriori ha formattato</pre>                                                                                                                                                                                                                                                                                                                                                                                                                                                                                                                                                                                                                                                                                                                                                                                                                                                                                                                                                                                                                                                                                                                                                                                                                                                                                                                                                                                                                                                                                                                                                                                                                                                                                                                                                                                                                                                                       | (<br>(    |
|                  | quantification of the uncertainties assumeshould have th<br>ha formattato: Evidenziato<br>ha eliminato: /assessment/impact<br>ha formattato: Evidenziato<br>ha eliminato: as for example<br>ha formattato<br>ha eliminato: to and a-posteriori<br>ha formattato<br>ha eliminato: ).                                                                                                                                                                                                                                                                                                                                                                                                                                                                                                                                                                                                                                                                                                                                                                                                                                                                                                                                                                                                                                                                                                                                                                                                                                                                                                                                                                                                                                                                                                                                                                                                                                                                                                                                                                                                                                         | (<br><br> |
|                  | quantification of the uncertainties assumeshould have th<br>ha formattato: Evidenziato<br>ha eliminato: /assessment/impact<br>ha formattato: Evidenziato<br>ha eliminato: as for example<br>ha formattato<br>ha eliminato: to and a-posteriori<br>ha formattato<br>ha eliminato: ).<br>ha eliminato: Internter-comparison studies ( <i>Choulge</i> )                                                                                                                                                                                                                                                                                                                                                                                                                                                                                                                                                                                                                                                                                                                                                                                                                                                                                                                                                                                                                                                                                                                                                                                                                                                                                                                                                                                                                                                                                                                                                                                                                                                                                                                                                                        | ę         |
|                  | <pre>quantification of the uncertainties assumeshould have th ha formattato: Evidenziato ha eliminato: /assessment/impact ha formattato: Evidenziato ha eliminato: as for example ha formattato ha eliminato: to and a-posteriori ha formattato ha eliminato: ). ha eliminato: Internter-comparison studies (Choulge ha eliminato: Assessingssessing the feasible potenti </pre>                                                                                                                                                                                                                                                                                                                                                                                                                                                                                                                                                                                                                                                                                                                                                                                                                                                                                                                                                                                                                                                                                                                                                                                                                                                                                                                                                                                                                                                                                                                                                                                                                                                                                                                                            |           |
|                  | <pre>quantification of the uncertainties assumeshould have th ha formattato: Evidenziato ha eliminato: /assessment/impact ha formattato: Evidenziato ha eliminato: as for example ha formattato ha eliminato: to and a-posteriori ha formattato ha eliminato: ). ha eliminato: Internter-comparison studies (Choulge ha eliminato: Assessingssessing the feasible potenti ha eliminato: EFmission factors (EFs) to any</pre>                                                                                                                                                                                                                                                                                                                                                                                                                                                                                                                                                                                                                                                                                                                                                                                                                                                                                                                                                                                                                                                                                                                                                                                                                                                                                                                                                                                                                                                                                                                                                                                                                                                                                                |           |
|                  | <pre>quantification of the uncertainties assumeshould have th ha formattato: Evidenziato ha eliminato: /assessment/impact ha formattato: Evidenziato ha eliminato: as for example ha formattato ha eliminato: to and a-posteriori ha formattato ha eliminato: ). ha eliminato: Internter-comparison studies (Choulge ha eliminato: Assessingssessing the feasible potenti ha eliminato: EFmission factors (EFs) to any ha eliminato: .</pre>                                                                                                                                                                                                                                                                                                                                                                                                                                                                                                                                                                                                                                                                                                                                                                                                                                                                                                                                                                                                                                                                                                                                                                                                                                                                                                                                                                                                                                                                                                                                                                                                                                                                                |           |
|                  | <pre>quantification of the uncertainties assumeshould have th ha formattato: Evidenziato ha eliminato: /assessment/impact ha formattato: Evidenziato ha eliminato: as for example ha formattato ha eliminato: to and a-posteriori ha formattato ha eliminato: ). ha eliminato: Internter-comparison studies (Choulge ha eliminato: EFmission factors (EFs) to any ha eliminato: . ha eliminato: . ha eliminato: . ha eliminato: . </pre>                                                                                                                                                                                                                                                                                                                                                                                                                                                                                                                                                                                                                                                                                                                                                                                                                                                                                                                                                                                                                                                                                                                                                                                                                                                                                                                                                                                                                                                                                                                                                                                                                                                                                    |           |
|                  | <pre>quantification of the uncertainties assumeshould have th ha formattato: Evidenziato ha eliminato: /assessment/impact ha formattato: Evidenziato ha eliminato: as for example ha formattato ha eliminato: to and a-posteriori ha formattato ha eliminato: ]. ha eliminato: Internter-comparison studies (Choulge ha eliminato: Assessingssessing the feasible potenti ha eliminato: EFmission factors (EFs) to any ha eliminato: . ha eliminato: . ha eliminato: nterelying on a large spectrum of ha formattato</pre>                                                                                                                                                                                                                                                                                                                                                                                                                                                                                                                                                                                                                                                                                                                                                                                                                                                                                                                                                                                                                                                                                                                                                                                                                                                                                                                                                                                                                                                                                                                                                                                                  |           |
|                  | <pre>quantification of the uncertainties assumeshould have th ha formattato: Evidenziato ha eliminato: /assessment/impact ha formattato: Evidenziato ha eliminato: as for example ha formattato ha eliminato: to and a-posteriori ha formattato ha eliminato: Internter-comparison studies (Choulge ha eliminato: Assessingssessing the feasible potenti ha eliminato: EFmission factors (EFs) to any ha eliminato: relayingelying on a large spectrum of ha formattato ha eliminato: is an</pre>                                                                                                                                                                                                                                                                                                                                                                                                                                                                                                                                                                                                                                                                                                                                                                                                                                                                                                                                                                                                                                                                                                                                                                                                                                                                                                                                                                                                                                                                                                                                                                                                                           |           |
|                  | <pre>quantification of the uncertainties assumeshould have th ha formattato: Evidenziato ha eliminato: /assessment/impact ha formattato ha eliminato: as for example ha formattato ha eliminato: to and a-posteriori ha formattato ha eliminato: ). ha eliminato: Internter-comparison studies (Choulge ha eliminato: Assessingssessing the feasible potenti ha eliminato: EFmission factors (EFs) to any ha eliminato: . ha eliminato: nelayingelying on a large spectrum of ha formattato ha eliminato: is an ha formattato ha eliminato: Evidenziato</pre>                                                                                                                                                                                                                                                                                                                                                                                                                                                                                                                                                                                                                                                                                                                                                                                                                                                                                                                                                                                                                                                                                                                                                                                                                                                                                                                                                                                                                                                                                                                                                               |           |
|                  | <pre>quantification of the uncertainties assumeshould have th ha formattato: Evidenziato ha eliminato: /assessment/impact ha formattato ha eliminato: as for example ha formattato ha eliminato: to and a-posteriori ha formattato ha eliminato: ). ha eliminato: Internter-comparison studies (Choulge ha eliminato: Assessingssessing the feasible potenti ha eliminato: EFmission factors (EFs) to any ha eliminato: . ha eliminato: relayingelying on a large spectrum of ha formattato ha eliminato: is an ha formattato ha eliminato: component of</pre>                                                                                                                                                                                                                                                                                                                                                                                                                                                                                                                                                                                                                                                                                                                                                                                                                                                                                                                                                                                                                                                                                                                                                                                                                                                                                                                                                                                                                                                                                                                                                              |           |
|                  | <pre>quantification of the uncertainties assumeshould have th ha formattato: Evidenziato ha eliminato: /assessment/impact ha formattato: Evidenziato ha eliminato: as for example ha formattato ha eliminato: to and a-posteriori ha formattato ha eliminato: ]. ha eliminato: Internter-comparison studies (Choulge ha eliminato: Assessingssessing the feasible potenti ha eliminato: EFmission factors (EFs) to any ha eliminato: . ha eliminato: is an ha formattato ha eliminato: is an ha formattato: Evidenziato ha eliminato: Evidenziato ha eliminato: Evidenziato ha eliminato: Evidenziato ha eliminato: Evidenziato ha formattato ha eliminato: Evidenziato ha formattato ha formattato: Evidenziato</pre>                                                                                                                                                                                                                                                                                                                                                                                                                                                                                                                                                                                                                                                                                                                                                                                                                                                                                                                                                                                                                                                                                                                                                                                                                                                                                                                                                                                                      |           |
|                  | <pre>quantification of the uncertainties assumeshould have th ha formattato: Evidenziato ha eliminato: /assessment/impact ha formattato: Evidenziato ha eliminato: as for example ha formattato ha eliminato: to and a-posteriori ha formattato ha eliminato: ]. ha eliminato: Internter-comparison studies (Choulge ha eliminato: Assessingssessing the feasible potenti ha eliminato: EFmission factors (EFs) to any ha eliminato: neliminato: [Interelying on a large spectrum of ha formattato ha eliminato: Evidenziato ha eliminato: enayingelying on a large spectrum of ha formattato ha eliminato: Is an ha formattato ha eliminato: Evidenziato ha eliminato: Evidenziato ha eliminato: enayingelying on a large spectrum of ha formattato ha eliminato: Is an ha formattato ha eliminato: Evidenziato ha eliminato: Evidenziato ha eliminato: enayingelying on a large spectrum of ha formattato ha eliminato: Is an ha formattato ha eliminato: Is an ha formattato: Evidenziato ha eliminato: enayingelying ha eliminato: enaying ha eliminato: Is an ha formattato: Evidenziato ha eliminato: enaying ha eliminato: Evidenziato ha elim</pre>                   |           |

Default EFs compiled by IPCC Guidelines (IPCC Guidelines, 2006, hereafter referred to as IPCC-06) 209 210 are adopted by EDGAR for most sources and countries, supplemented by information from scientific 211 literature, and other references for specific processes and/or countries. Janssens-Maenhout et al. (2019) 212 produced a detailed description of data providers and methodological choices for the GHGs emissions 213 of EDGAR. Further information on methodological aspects of data collection and sources are given by 214 *Crippa et al. (2020a).* 215 This study addresses the uncertainty of the anthropogenic sources covered by EDGAR, which might be 216 not exhaustive. Therefore, nothing can be said about the uncertainty stemming from source categories 217 not currently encompassed within the inventory (e.g., fugitive CO<sub>2</sub> from low temperature oxidation of 218 coal mines, fugitive CH<sub>4</sub> from managed wetlands, N<sub>2</sub>O from crab ponds as part of aquaculture). 219 Uncertainty assessment of spatially distributed sources (emission gridmaps) is not within the scopes of 220 this study.

# 220

# 222 2.1 Emissions and <u>Their Uncertainties</u>

The uncertainty of AD (u<sub>AD</sub>) collected by international agencies or organisations (e.g. the Food and Agriculture Organization (FAO), International Energy Agency (IEA)) is of statistical nature, stemming from incompleteness, representativeness of sampling, imputation of missing data, extrapolation (e.g. projecting to future years) (*Rypdal and Winiwarter, 2001; Olivier, 2002; IPCC-06*). Other aspects to take into <u>consideration</u> when compiling a global inventory are the degree of wealth of a country as well as the year under study. Less developed countries and countries whose economy has fully developed in recent years, are more probable to have not yet developed a reliable statistical system. <u>Similarly, AD of</u>

countries with transitional economies are expected to be more accurate for recent years (*Janssens- Maenhout et al.*, 2019).

Uncertainty in EF (u<sub>EF</sub>) has many sources, as for instance: degree of representativeness of the limited 233 number of observations underlying the EF, for the activity that is addressed, including under-234 representativity of operating conditions; inaccuracy of assumptions and/or of source aggregation (e.g. 235 assumption of constancy in time); bias, variability and/or random errors. Due to the non-statistical 236 nature of u<sub>EF</sub>, its quantification eludes a general methodological approach. IPCC adopts a tiered 237 approach for estimating uncertainty, accounting for different levels of sophistication (IPCC-06). Tier 1 238 uncertainties on default EFs are based on expert judgement, which often offers a range of uncertainties 239 for a given process, source, and/or fuel. Higher tiers (up to Tier 3) offer more elaborate estimates, based on localized measurements/ad-hoc experiments on specific emission factors and for specific processes. 240 241 Further, the model used to build emission inventories based on activity statistics may be too simplified 242 (e.g., based on linearization and/or linear regression due for example to poor understanding, lack of 243 data, etc.), and may not fully capture the complexity of a given emission process. These 'model' errors 244 are difficult to be assessed in isolation from other sources of uncertainty, and are generally attributed to 245 uncertainties in EFs (Rypdal and Winiwarter 2001; Cullen and Frey, 1999). This study reflects the methodological approach of EDGAR adopting default EFs, thus associated with 246

This study reflects the inclusion of proven of provide a provide a study reflects the inclusion of provide a study as in similar ones (*Rypdal and* Winiwarter, 2001; Olivier, 2002; Janssens-Maenhout et al., 2019), is used in a rather broad sense, lumping together all mentioned sources of errors due to current limited knowledge to distinguish among them. After IPCC introduced quantitative uncertainty in GHG inventories, the inventory uncertainty is usually expressed as two standard deviations, approximately corresponding to 95% confidence for a variable with a normal distribution (i.e., the uncertainty reflects the square root of the variance of the variable, multiplied by a coverage factor of 2 to provide a confidence interval of 95%).

# ha eliminato: sources ha eliminato: process ha eliminato: is

ha eliminato: UNCERTAINTY

| ha eliminato: and Peters                                                                                   |
|------------------------------------------------------------------------------------------------------------|
| ha eliminato: considerations                                                                               |
| ha formattato: Evidenziato                                                                                 |
| ha eliminato: for past decades                                                                             |
| ha formattato: Evidenziato                                                                                 |
| ha eliminato: expectedly less                                                                              |
| ha formattato: Evidenziato                                                                                 |
| ha eliminato: than                                                                                         |
| ha formattato: Evidenziato                                                                                 |
| ha formattato: Evidenziato                                                                                 |
| ha eliminato: inexactness                                                                                  |
| ha formattato: Evidenziato                                                                                 |
| <b>ha eliminato:</b> (e.g. due to measurement errors); under-<br>representativity of operating conditions. |
| ha eliminato: judgements                                                                                   |
| ha eliminato: Accuracy of the uncertainty estimate increases with tier.                                    |
| ha eliminato: .                                                                                            |
| ha eliminato:                                                                                              |
| ha eliminato: , e.g.,                                                                                      |
| ha eliminato: ),                                                                                           |
| ha eliminato: assess                                                                                       |
| ha eliminato: source                                                                                       |
| ha eliminato: EF factors                                                                                   |
| ha eliminato: Tier1                                                                                        |
| ha eliminato:                                                                                              |
| ha eliminato: and Peters                                                                                   |
| ha eliminato:                                                                                              |

Finally, the uncertainty tackled here shall not be confused with the variability stemming from a range

- (or ensemble) of estimates. The variability is used as proxy of structural uncertainty in the faith that a
- range of models using diverse underlying assumptions would span the true uncertainty space. However, the estimates are seldom 'diverse' as they build up from same data/assumptions (sometimes different
- versions of the same model are used) leading to overconfident estimates (*Solazzo et al.*, 2018).

# 285 2.1.1 UNCERTAINTY IN ACTIVITY DATA

Table 1 summarizes the uncertainty for AD. When two values are listed (e.g.  $\pm 5\%$ ;  $\pm 10\%$ ), the lower uncertainty value (i.e.  $\pm 5\%$ ) is assigned to countries with developed economy, while the larger values

(i.e.  $\pm 10\%$ ) to countries with less developed economy or with economy in transition.

# 289 TABLE 1.

According to IPCC-06, u<sub>AD</sub> for fuel combustion activities (mostly derived from IEA statistics) are estimated with high confidence (5 to 10% uncertainty). The same uncertainty range is estimated for

fugitive emissions (referring to venting and flaring during oil and gas production). u<sub>AD</sub> in the residential

(10 to 20%) and <u>in the</u> aviation and navigation (5 to 25%) sectors are assumed more conservative, to

account for the under-representativeness of the sample and for the difficulty of distinguishing between

domestic and international fuel consumption (IPCC-06). For combustion processes using biofuels, the

statistics is less robust. *Olivier* (2002) suggests u<sub>AD</sub> of 30% for industrialised countries and 80% for less

developed ones (based on IPCC-06 recommendations). Recent updates (*Andreae*, 2019) confirm theseestimates.

Uncertainty for some chemistry production processes and waste is calculated on the total emission rather than on AD and EF separately, and is discussed later. The waste sector also utilizes a slightly more elaborated emission estimate model than the simple multiplication of AD and EF. It assumes that emissions are not instantly released into the atmosphere, but <u>are accumulated</u> and continue to emit even several years after their disposal. The model for the waste sector depends on several parameters and assumptions (detailed in section 3.1.5).

2.1.2 UNCERTAINTY ON EMISSIONS FACTORS

Tables 2 and 3 define the uncertainties of EFs for CO<sub>2</sub>, and for CH<sub>4</sub> and N<sub>2</sub>O, respectively. Uncertainty
 of EFs for CO<sub>2</sub> is determined by the carbon content of the fuel and is relatively smaller and determined
 with higher level of accuracy than uncertainty of EFs for CH<sub>4</sub> and N<sub>2</sub>O. Moreover, <u>ther</u> for CH<sub>4</sub> and N<sub>2</sub>O
 lumps several sources of uncertainties, as mentioned earlier.

TABLE 2.

TABLE 3.

As <u>adverted before</u>, u<sub>EF</sub> are <u>founded</u> on Tier 1 estimates by IPCC-06, which are based on expert

judgments and, as such, they vary over wide ranges to account for a variety of conditions, For instance,

 $u_{EF}$  for N<sub>2</sub>O (agriculture and energy sources in particular) clearly <u>reflect</u> the large temporal variability and spatial heterogeneity of these processes.

2.2 EMISSION AGGREGATION AND UNCERTAINTY PROPAGATION

The vast majority of EFs in EDGAR are based on IPCC Tier 1 estimates (especially for combustion sources) to ensure:

- *completeness* accomplished through the inclusion of all relevant sources for a given year;

# ha eliminato: ; ha formattato: Evidenziato ha formattato: Evidenziato ha formattato: Evidenziato

| - | ha eliminato:                          |
|---|----------------------------------------|
| 1 | ha eliminato:                          |
| 1 | ha eliminato:                          |
| ľ | ha eliminato:                          |
| Ì | ha eliminato:                          |
| Ì | ha eliminato:                          |
| Ì | ha eliminato: however                  |
|   | ha eliminato: et al                    |
| ١ | ha eliminato: for estimating emissions |
| 1 | ha eliminato: they accumulate          |
|   | ha eliminato: also                     |
| Y | ha eliminato: ,                        |

ha eliminato: uncertainty of EFs

|   | ha eliminato: mentioned earlier                                             |
|---|-----------------------------------------------------------------------------|
| λ | ha eliminato: based                                                         |
| - | ha eliminato: provided                                                      |
|   | ha eliminato: ,                                                             |
| Η | ha eliminato:                                                               |
|   | ha eliminato: , e.g.                                                        |
| Ϊ | ha eliminato: reflects                                                      |
| / | ha eliminato: so                                                            |
| - | ha formattato: Evidenziato                                                  |
|   | <b>ha eliminato:</b> comparability, consistency, and transparency, allowing |
|   | ha formattato: Evidenziato                                                  |

consistency implying that the same methodology is applied through years for a given source; 344 345 comparability, assuring that emissions are comparable across countries, e.g. source definitions, emission calculations and emissions factors are the same across countries. 346 347 The adoption of comparable methods for source emissions and consistency implies that the uncertainties 348 of the final emission estimates are inter-dependent, as they stem from the same methodology. When emissions are combined/aggregated, this lack of independence cannot be neglected, and the following 349 assumptions are made: 350 351 a) emissions uncertainty (u<sub>EMI</sub>) is the sum of the squares of the uncertainty of AD (u<sub>AD</sub>) and the 352 uncertainty of EF ( $u_{EF}$ ), see Eq. 1); 353 b) <u>uncertainties</u> of different source categories are uncorrelated (e.g. waste and agriculture);

- c) <u>subsectors</u> of a given emission category for CH<sub>4</sub> and N<sub>2</sub>O are fully correlated, thus the uncertainty of the sum is the sum of the uncertainties;
- d) <u>when</u> dealing with CO<sub>2</sub>, full correlation is assumed for energy combustion sources sharing the same emission factor (fuel-dependent);
- aggregated emissions from same categories but different countries are assumed to be fully correlated, unless the emission factor is country-specific, or derived from higher tiers (i.e. emissions are not derived from default EF defined by IPCC but are retrieved by other sources and are specific to that country/process);
- f) <u>when</u> uncertainty is provided as a range (e.g. for the energy sector, IPCC-06 recommend that the <u>CH<sub>4</sub> EFs</u> are treated with an uncertainty ranging from 50% to 150%), the upper bound of the range is assigned to countries with less developed statistical infrastructure and <u>the lower</u> one to countries with more robust statistical infrastructure.
- Conditions a) and b) match the suggestion of the uncertainty chapter of the IPCC guidelines (IPCC06, Chapter 3), whilst the latter two conditions are more cautious formulations of the error
  propagation to account for covariances. More explicitly the uncertainty of the emission, u<sub>EMI</sub>, due
- to multiplying AD by EF is calculated as:

$$u_{\rm EMI} = \sqrt{(u_{EF}^2 + u_{AD}^2)}$$

The uncertainty on the emission,  $u_{EMI}$ , due to adding emissions is calculated as:

$$u_{EMI} = \frac{\sqrt{\sum_{i} (EMI, i * u_{EMI,i})^2}}{\sum_{i} |EMI, i|}$$

363

That is, basically, the squared sum of the uncertainty of each emission process normalised by the sum of emissions, which assumes that all emission sources are uncorrelated (IPCC-06). However, in general, the variance of the sum of any two terms  $x_1$  and  $x_2$  having variances of  $\sigma_1$  and  $\sigma_2$  is  $\sigma_{sum}^2 = \sigma_1^2 + \sigma_2^2 + 2cov(x_1, x_2)$ . Since the covariance can be expressed as  $2cov(x_1, x_2) = 2r\sigma_1 \sigma_2$ , where *r* is the coefficient of correlation, when r = 1 (full correlation), the variance of the sum becomes the linear sum of the two variances:

$$\sigma_{sum} = \underbrace{\sigma_1 + \sigma_2}_{correlated r=1} \ge \underbrace{\sqrt{\sigma_1^2 + \sigma_2^2}}_{uncorrelated r=0}$$

| - | ha eliminato: emission   | J |
|---|--------------------------|---|
|   |                          |   |
| - | ha eliminato: factors in | ۱ |

| ha eliminato: ) (Eq. 1      |  |
|-----------------------------|--|
| ha eliminato: Uncertainties |  |
| ha eliminato: Subsectors    |  |
| ha eliminato: When          |  |
| na eliminato: when          |  |

| ha eliminato: | Aggregated |
|---------------|------------|
|---------------|------------|

| ha eliminato: When                     |  |
|----------------------------------------|--|
| ha eliminato: methane emission factors |  |
| ha formattato: Evidenziato             |  |
| ha eliminato: upper                    |  |

| EQ. 1 🔸 | Formattato: Allineato a destra |
|---------|--------------------------------|
|         |                                |
|         | ha eliminato: ;                |
| EQ. 2 🔸 | <br>ha eliminato:              |

| Formattata  | Allineate | a doctra |
|-------------|-----------|----------|
| Formattato: | Allineato | a destra |

 ha formattato: Inglese (Irlanda)

 ha eliminato:
  $\sigma_1 + \sigma_2$  

 correlated r=1

5

EQ. 3 <

Therefore, for fully correlated variables, the uncertainty of their sum is simply the sum of their 394 uncertainties.

When uncertainties are larger than 100%, Eq. 2) tends to underestimate the uncertainty and a correction factor F<sub>C</sub> is recommended (IPCC-06), so that the uncertainty on the emission is: 396

$$u_{EMI,C} = u_{EMI,c} = u_{EMI,c} F_C$$

$$F_C = \left[\frac{(-0.72 + 1.0921u_{EMI} - 1.63x10^{-3}u_{EMI}^2 + 1.11x10^{-5}u_{EMI}^3)}{u_{EMI}}\right]^2$$
EQ. 4

where  $u_{EML,C}$  is the correction to be applied to the uncertainty estimated from error propagation. Eq. 4) 399 is used for multiplicative or quotient terms in the range u<sub>EMI</sub>∈[100%,230%] (Equation 3.3, IPCC-06 400 Volume 1 Chapter 3). The effect of Fc is to return larger uncertainties (see e.g. Choulga et al., 2020). 401 The use of  $F_C$  is based on the work by *Frey* (2003) to account for the error introduced in the 402 approximation of the analytical method compared to a fully numerical one (based on Monte Carlo 403 analysis). The error in the approximation increases with the uncertainty, and thus the correction factor 404 F<sub>C</sub> is needed, when dealing with large uncertainties, (Frey, 2003). The analysis presented in this study 405 takes into account for the correction factor F<sub>C</sub> (unless specifically indicated) and for simplicity the 'C' 406 is dropped in u<sub>EMI,C</sub> to yield u<sub>EMI</sub>.

This study assumes that uncertainties are normally distributed, unless specifically indicated by IPCC-408 06. The distribution is transformed to log-normal after the aggregation to avoid that the emissions take 409 negative, unphysical values when uncertainty is large. Hence, the probability distribution function 410 (PDF) is transformed to lognormal with the upper and lower uncertainty range defined according to 411 IPCC-06:

$$u_{EMI} = \frac{1}{EMI} \left( \exp(\ln(\mu_g) \pm 1.96 \ln(\sigma_g)) \right) - 1$$

where  $\mu_g$  and  $\sigma_g$  are the geometric mean and geometric standard deviation about EMI, the mean 413 emission 414

According to IPCC-06, the contribution to variance, var, share, of a specific emission process s emitting 415 EMIs, to the uncertainty of the total emissions EMItot is calculated as: 416

$$var share_s = \frac{u_{EMI,s}^2 * EMI_s^2}{EMI_{tot}^2}$$
 EQ. 6

#### 418 2.2.1 ADDITIONAL REMARKS

The assumption of correlation between subcategories (or fuel for energy sector emitting CO<sub>2</sub>) an 420 between countries for the same category (or fuel for energy-CO<sub>2</sub>) is introduced to ensure that th 421 uncertainty of sources sharing the same methodology for estimating the EF is propagated in case of 422 aggregation. If the same methodology is applied to estimate the emission for a given category and for 423 group of countries, then the correlation is kept when calculating the total emission of that group of 424 countries for that category. Similar assumptions were adopted by e.g., Bond et al. (2004) and 425 Bergamaschi et al. (2015) (though for different inventories). This is a direct implication of the 426 consistency and cross-country comparability of EDGAR, that adopts Tier 1 EFs defined by IPCC-06

|---|--------------------------------|
|   |                                |
|   |                                |

| - | ha eliminato: Where               |
|---|-----------------------------------|
| + | ha eliminato:                     |
|   | ha formattato: Non Apice / Pedice |
| 4 | ha eliminato: Fc                  |

| _ | ha eliminato: , according to Frey (2003), for |
|---|-----------------------------------------------|
|   | ha eliminato: .                               |
|   | ha eliminato: indicted                        |
| _ | ha eliminato: To                              |

| - | ha eliminato: Eq                                  |
|---|---------------------------------------------------|
|   |                                                   |
| - | ha eliminato: (                                   |
| 1 | ha eliminato: ) EMI.                              |
| Ν | ha formattato: Tipo di carattaro: Carsiva Inglasa |

|    | (Regno Unito)                       |
|----|-------------------------------------|
|    | ha eliminato: The                   |
|    | ha eliminato: share                 |
| d  | ha eliminato: according to IPCC-06. |
| ie |                                     |
| of | <br>ha eliminato: propagate         |
| a  | ha eliminato: that                  |

ha eliminato: These same

ha eliminato: emission factors

EQ. 5 🖪

for most of the inventory. By contrast, if each country follows diverse methods to estimate the EFs for a given source category,  $\mu_{EF}$  stemming from that methodology does not co-vary when calculating the total of that category, and thus Eq. 2) holds. Some further considerations:

- the assumption of source/country correlation is the main difference between the uncertainty
   estimated in this study and the uncertainty reported by, e.g., *Petrescu et al. (2020)* for
   EU27+UK, where no correlation was assumed, although not all countries developed
   independent methods to estimate EFs;
- the choice of assuming 'full' correlation (i.e. correlation coefficient of one) is conservative in
   the sense that it <u>returns</u> the upper bound of u<sub>EMIs</sub> and is motivated by two main reasons: it
   simplifies the calculation (see Eq. 3)), and there are no indications how to better estimate r;
- EDGAR does include country-specific EFs for some processes and countries. Those are 456 retrieved from the scientific literature or derived from technical collaborations, and through the 457 continuous updates over the last two decades (e.g. EFs for cement production are computed 458 including information on country-specific clinker fractions. EFs for landfills consider the 459 country specific waste composition and recovery; EFs for enteric fermentation of cattle include 460 country/region specific information on milk yield, carcass weight and many other parameters, 461 etc.). These instances are flagged in our methodology, and the uEF is not propagated when 462 aggregating these sources.
- 3, UNCERTAINTY IN EMISSION SECTORS
- 3.1 Emissions from CO<sub>2</sub>, CH<sub>4</sub> and N<sub>2</sub>O
- 3.1.1 POWER INDUSTRY SECTOR

IPCC sector 1.A includes the EDGAR categories related to combustion of fossil and biofuels for energy
production (ENE), manufacturing (IND), energy for buildings (RCO), oil refineries and transformation
industry (REF, TRF), aviation (TNR aviation), shipping (TNR ship), and road transport (TRO).
Emissions from biofuel burning (e.g. wood) in sector 1.A are considered carbon neutral and are

- calculated for  $CH_4$  and  $N_2O$  only.
- EDGAR adopts AD statistics of fossil fuel combustion compiled by the IEA (IEA, 2017) for developed
- and developing countries, integrated with data from EIA (2018) for biofuels.

# 473 TABLE 4.

The share of GHGs emissions from industrialised and developing countries is reported in Table 4 to aid

later interpretation of the uncertainty shares. In fact, in countries with developed economy (Table S 1)

energy statistics <u>are</u> considered <u>to have</u> lower uncertainty than <u>in</u> countries in development (*Olivier* 

2002). IPCC suggests  $u_{AD}$  for the power industry ranging between 5 to 10%. We have assigned 5% to

industrialised countries and 10% uncertainty to developing countries to account for less robust census

- capability. IPCC-06 provides fuel-dependent  $u_{EF}$  for CO<sub>2</sub> (**Table 2**), which have been mapped to match 480 the fuels in each EDGAR emission category.  $u_{EF}$  for CO<sub>2</sub> are relatively small as reflected by the (well 481 known) carbon content of the fuel.
- For CH<sub>4</sub> and N<sub>2</sub>O, EFs are more uncertain than for CO<sub>2</sub>. IPCC-06 suggests a wide range of  $u_{EF}$  for the 483 whole energy sector, ranging between 50% and 150% for CH<sub>4</sub> and between one tenth and ten times the
- mean emission value for  $N_2O$ . These estimates are provided by expert judgement based on the reliability
- of current estimates. The reasons for such high uncertainty are those mentioned before, <u>i.e.</u>, lack of
- understanding of emission processes and of relevant measurements, uncertainty in measurements, poor

ha eliminato: the uncertainty

### ha eliminato: The

| ha eliminato: .          |
|--------------------------|
| ha eliminato: The        |
| ha eliminato: will yield |
| ha eliminato: (          |
| ha eliminato: )          |
| ha eliminato: as         |
| ha eliminato: derived    |
| ha eliminato: through    |
| ha eliminato: in         |
| ha eliminato: ,          |
| ha eliminato: ,          |
| ha eliminato: which      |
| ha eliminato: .          |

ha eliminato:

| -                 | ha eliminato: Table 5    |
|-------------------|--------------------------|
| -                 | ha eliminato: is         |
| T                 | ha eliminato: having     |
| ľ                 | ha eliminato: for        |
| $\langle \rangle$ | ha eliminato: and Peters |
|                   | ha eliminato:            |
|                   | ha eliminato: Table      |
| -                 | ha eliminato: TABLE 5.¶  |
|                   |                          |

| - | ha eliminato: that is | J |
|---|-----------------------|---|
| - | ha eliminato: lack    | ١ |

representativeness of the full range of operating conditions. EFs for biofuels combustion are highly

uncertain, estimated in the range 30% (<u>Andreae</u> and Merlet, 2001) to 80% (Olivier, 2002). Recently,

Andreae (2019) has reviewed  $u_{EF}$  to less than 20% (6-18% for CH<sub>4</sub> from the major burning categories

<u>savannah</u>, forests, and biofuel). The uncertainty of processes using biofuels is calculated separately and

then combined with the fossil fuel uncertainty, assuming no correlation, see Eq. 2).

# 518 FIGURE 1.

Emissions of  $CO_2$  account for over 90% of world's total GHG emissions from fuel combustion, and are assessed with high degree of confidence (Figure 1a,b,c) due to the accuracy of  $u_{EF}$  reflecting the carbon

content of the fuel. Thus, the share of emission for each subcategory (manufacturing, transformation

and power industry, oil refinery, residential heating, road and non-road transport) is mirrored by the

share each category contributes to the sector uncertainty (Figure 2), although with some notable

exceptions for non-road transport in Brazil (large share of highly uncertain domestic aviation and inland

water shipping), and transformation industry in Russia (share of emission and uncertainty of ~10% and

$\sim$  37%, respectively).

# 527 FIGURE 2.

The very low confidence in  $N_2O$  emissions is responsible for almost 50% of world's total uncertainty

- (Figure 1f) although N<sub>2</sub>O only accounts for a minor portion of total emissions in this sector (less than
- 1%). According to, e.g., *Lee et al. (2013)*, the suggested IPCC-06 uncertainty on power plant emission
   of N<sub>2</sub>O might be too high (the authors report a range of -11.43% and +12.86% for combined-cycle
- power plant in Korea). An alternative  $u_{EF}$  estimation for N<sub>2</sub>O in the fossil fuel combustion sector is set
- in the range  $\pm 50\%$  (developed countries) to  $\pm 150\%$  (countries with economy in development). This
- choice also reflects previous uncertainty estimation by *Olivier (2002)*.
- The N<sub>2</sub>O emission uncertainty and the N<sub>2</sub>O contribution to uncertainty in sector 1.A become as shown in Figure 3.

# 537 FIGURE 3.

The uncertainty distribution (Figure 3) and relative contribution <u>reflect</u> the weight of the component

GHGs and the world's total uncertainty (10%) is only slightly larger than the uncertainty of CO<sub>2</sub> (7%,

Figure 1a,b,c). Adopting the u<sub>EF</sub> of 50-150% for N<sub>2</sub>O in sector 1.A reflects the large uncertainty

associated with this sector and allow comparability/aggregation with other gases (Figure 3b).

3.1.2 FUGITIVE EMISSIONS FROM COAL, OIL AND NATURAL GAS

Fugitive emissions from solid fuels (mainly coal, 1.B.1) and from oil and natural gas (1.B.2) are covered

by the EDGAR's categories REF, TRF and by fuel exploitation PRO. As pointed out in *IPCC-06*,

- uncertainty in the fugitive emissions sector arises from applying the same EF to all countries (Tier 1
- approach) and from uncertainty in the emission factors themselves.

AD for coal statistics is a collection of products (full details are provided by *Janssens-Maenhout et al.* 

- (2019) and references therein): the World Coal Association (2016); JEA (2017) for exploration of gas
- and oil; UNFCCC (2018) and CIA (2016) for transmission and distribution; IEA (2017) for venting and
- flaring, complemented with data from GGFR/NOAA data (2019) and Andres et al. (2014). According
- to Olivier, (2002),  $u_{AD}$  for sector 1.B lies within the range ±5 to ±10%, which is aligned with the
- estimates provided by *IPCC-06*.

| ha eliminato: representativity            |
|-------------------------------------------|
| ha eliminato: operational                 |
| ha eliminato: Andeae                      |
| ha formattato: Tipo di carattere: Corsivo |
| ha eliminato: et al.,                     |
| ha formattato: Tipo di carattere: Corsivo |
| ha eliminato: savanna                     |
| ha eliminato: uncorrelation (             |
|                                           |

| ha eliminato: |  |
|---------------|--|
|               |  |
|               |  |

| -{ | ha eliminato: Fig          |
|----|----------------------------|
|    |                            |
| -{ | ha formattato: Evidenziato |
| -{ | ha formattato: Evidenziato |
| -( | ha eliminato: et al.       |
| 7  | ha formattato: Evidenziato |
| -{ | ha eliminato: :            |

ha eliminato:

#### ha eliminato:

| ha formattato: Tipo di carattere: Corsivo     |
|-----------------------------------------------|
| ha formattato: Tipo di carattere: Non Corsivo |
| ha formattato: Tipo di carattere: Corsivo     |
| ha formattato: Tipo di carattere: Corsivo     |
| ha eliminato: at al.                          |

Fugitive emissions from solid fuels (1.B.1) in EDGARv4 and v5 are dealt with by considering emission

factors from *IPCC*, 06, supplemented with *EMEP/EEA* (2013) Guidebook for coal and UNFCCC

(2018). For oil and natural gas (<u>1.B.2</u>), we use information from the <u>IPCC\_06</u>, supplemented with data

of *UNFCCC (2014)*. While gas transmission through large pipelines is characterised with relatively 571 small country-specific emission factors of *Lelieveld et al. (2005)*, much larger and material dependent

shall could y specific consistent actors of *Lenevera et al.* (2005), international actors of

based on country-specific UNFCCC (2014) data for reporting countries (and the average value as

default for all other countries) (*Janssens-Maenhout et al.*, 2019).

*IPCC-06* provides a detailed synthesis of uncertainty associated with EFs for sectors 1.B.1 and 1.B.2, 576 distinguishing between developing and developed countries (Tables 4.2.4 and 4.2.5 of *IPCC-06*, chapter 577 4).  $u_{EF}$  is the same for CO<sub>2</sub> and CH<sub>4</sub>, while is larger for N<sub>2</sub>O. A summary of uncertainty ranges is 578 provided in Table 3.

Uncertainties in the 1.B.1 sector depend on the type of mining activity: 'surface' (surf), 'underground' (und) and 'abandoned' (abandon).  $u_{\rm EF}$  for these sectors can be rather large (>100%), as detailed in 580 581 Table 3, according to IPCC-06 and in line with Olivier (2002). For 1.B.2, the distinction is made 582 between leakage in production (prod), transmission and distribution (trans), and venting/flaring (vent). 583 The uncertainty is estimated as large as three times the average emission value for some instances (Table 584 3) for  $CH_4$  and  $CO_2$  and up to 1000% for flaring  $N_2O$  emission. We note that while some AD are known 585 or retrievable through various governmental agencies (e.g. number of gas production wells, miles of 586 pipelines, number of gas processing plants), other activity data (e.g., storage tank throughput, number 587 of various types of pneumatic controllers, and reciprocating engines are more uncertain. As reported 588 by EPA, 'petroleum and gas infrastructure consist of millions of distinct emission sources, making

measurement of emissions from every source and component practically unfeasible' (EPA, 2017).

#### 590 **FIGURE 4.**

The <u>fugitive emission</u> sector is dominated by  $CH_4$  emissions and this is reflected in the contribution to the total uncertainty of GHG emission from sector 1.B (Figure 4e). The upper world uncertainty estimate exceeds 110%, almost entirely due to  $CH_4$  emissions. For the <u>USA</u>, upper uncertainty <u>estimates</u> for oil and natural gas (Figure 4c) of 23% is <u>slightly less than the EPA's upper estimate of 30% for the</u> natural gas system (*EPA*, 2017) and that of *Littlefield et al.* (2017) of 29%, while for the petroleum system the EPA's uncertainty is much larger (149%), possibly due to higher  $u_{AD}$ .

The uncertainty of individual countries mirrors the distinction made between developed and developing countries, mostly visible for fugitive emissions from oil and natural gas (Figure 4c) but also in the detailed  $u_{EF}$  provided by *IPCC-06* for the various emitting stages of extraction, distribution, transport, and storage. The composition of emissions for the five top emitters in sector 1.B.2.b can be used to illustrate this aspect.

#### 602 TABLE 5.

The <u>USA</u> and Russia have country-specific EFs, which are defined for all stages of the fugitive emissions from natural gas, and therefore the accuracy is higher. Iran, Saudi Arabia, China have a very large share of emissions due to the production stage of natural gas (approximately 85%, 97%, 76%, respectively, **Table 5**), to which  $u_{EF} = \pm 75\%$  applies, and a much lower share of emissions apportioned to the other stages (i.e. transmission and distribution), approximately 10% due to gas distribution with an uncertainty of -40% to +500% (including the correction factor Eq. 4)), contributing to the very low confidence in the emission estimate shown in Figure 4e, compared with the medium confidence for

# ha eliminato: 1B1 ha eliminato: EDGAR Formattato: SpazioDopo: 10 pt, Interlinea: multipla 1.15 ri ha formattato: Tipo di carattere: Corsivo ha eliminato: (2006) Guidelines ha eliminato: 1B2 ha eliminato: UNFCCC as well as from ha formattato: Tipo di carattere: Corsivo ha eliminato: (2006) guidelines ha formattato: Tipo di carattere: Corsivo ha eliminato: (2006) guidelines ha eliminato: ¶ Formattato: Non regolare lo spazio tra testo asiatico e in alfabeto latino, Non regolare lo spazio tra testo asiatico e caratteri numerici ha formattato: Tipo di carattere: Corsivo ha formattato: Tipo di carattere: Corsivo Formattato: SpazioDopo: 10 pt, Interlinea: multipla 1.15 ri ha formattato: Tipo di carattere: Corsivo ha formattato: Citazione Carattere ha eliminato: et al. ha formattato: Citazione Carattere ha eliminato: ). ha formattato: Tipo di carattere: Corsivo ha formattato: Tipo di carattere: Corsivo ha eliminato: ¶ ha eliminato: US ha eliminato: estimate ha eliminato: in ha eliminato: country's ha eliminato: in this study ha eliminato: ha formattato: Tipo di carattere: Corsivo ha eliminato: 6 Formattato: Destro 0 cm ha eliminato: US ha eliminato: ha eliminato: ), ha eliminato: ) ha eliminato: ha eliminato: ).

USA and Russia, to which country-specific  $u_{EF}$  are applied ( $\pm 25\%$ ) (Table 3). The high uncertainty in the transmission/distribution sectors is the main <u>cause</u> for the difference in uncertainty apportionment.

Variability of bottom-up estimates of CH<sub>4</sub> emissions from coal mining (-29%, +43%) and natural gas 636 637 and oil systems (-16%, +15%), as recently reported by Saunois et al. (2020), stems from methodologies and parameters used, including emission factors, 'which are country- or even site-specific, and the few 638 field measurements available often combine oil and gas activities and remain largely unknown' 639 640 (Saunois et al., 2020). The authors reported examples of very large variability of EFs between 641 inventories, even of 2 orders of magnitude for oil production and by one order of magnitude for gas 642 production. Moreover, large uncertainties in emissions of CH4 from venting and flaring at oil and gas 643 extraction facilities were reported by e.g. Peischl et al. (2015). Gas distribution stage is a further large 644 source of uncertainty, in particular in countries with old gas distribution city networks using steel pipes 645 now distributing dry rather than wet gas, with potentially more leakages (Janssens-Maenhout et al., 646 2019). Analysis based on inversion modelling by Turner et al (2015) found, for the North America 647 region an error variability of -43% to 106% (with respect to the prior estimate based on EDGAR v4.2) 648 attributed to emissions from oil and gas. Hence, the uncertainty in Figure 4c might be too low for 649 industrialised countries. For completeness, we show an alternative application of uncertainty ranges for sector 1.B.2 (oil and gas), as suggest by Olivier (2002), assigning  $u_{AD} = \pm 5$  and  $\pm 15\%$  (industrialised 650 and developing countries, respectively) and  $u_{EF} = \pm 100\%$  to all countries and  $u_{EF}$  of 50% to countries 651 652 for which EF are specifically estimated (Tier 3).

#### 653 FIGURE 5.

The resulting distribution (Figure 5) reflects the comparable uncertainty of these emissions across countries. Global u<sub>EMI</sub> is of approximately 100%, thus slightly less than the uncertainty obtained by

applying the *IPCC-06* recommendations (122%, Figure 4e).

3.1.3 INDUSTRIAL PROCESSES AND PRODUCT USE (IPPU)

IPCC category 2 covers non-combustion emissions from industrial production of cement, iron and steel,

lime, soda ash, carbides, ammonia, methanol, ethylene, adipic and nitric acid and other chemicals and

660 the non-energy use of lubricants and waxes (Janssens-Maenhout et al., 2019). The EDGAR sectors

<u>CHE (production of chemicals)</u>, FOO (food production), PAP (paper and pulp production), IRO (iron

and steel), non-energy use of fuels (NEU), non-ferrous metal production (NFE) and non-metallic

minerals production (NMM) cover the industrial process emissions.

Activity statistics for industrial processes are retrieved from several reporting providers, as detailed by 664 Janssens-Maenhout et al., 2019, and Crippa et al, 2019, For this class of processes uAD are higher than 665 666  $u_{EF}$  due to the deficiency or incompleteness of country specific data and <u>reluctancy</u> by companies to 667 disclose production data. CO2 emissions in EDGAR are based on Tier 1 EF for clinker production, 668 whereas cement clinker production is calculated from cement production reported by USGS (2014). The fraction of clinker is based on data reported to UNFCCC for European countries, to the China Cement 669 670 Research Institute (www.ccement.com; yjy.ccement.com/) and the National Bureau Statistics of China 671 (for historic years) for China and to the 'getting the numbers right' for non-Annex I countries 672 (https://gccassociation.org/gnr/). According to IPCC-06, the uncertainty for cement production stems 673 prevalently from u<sub>AD</sub>, and to a lesser extent from u<sub>EF</sub> for clinker (*IPCC-06*, chapter 2). For Tier 1, the 674 major uncertainty component is the clinker fraction of the cement(s) produced and u<sub>AD</sub> can be as high 675 as 35%. We assume  $\mu_{EMI}$  of  $11\frac{6}{2}$  to 60% depending on the accuracy of clinker data.

As for cement, the  $u_{AD}$  for lime outweighs  $u_{EF}$  due to lack of country specific data. We assume  $u_{AD}$  of  $\pm 35\%$  and  $u_{EF} = \pm 3\%$ . For glass, glass production data are typically measured accurately as reflected by

ha eliminato: responsible

ha formattato: Pedice

| - | ha eliminato: A more realistic     |  |
|---|------------------------------------|--|
| - | ha eliminato: e.g.                 |  |
| 1 | ha eliminato: et al.               |  |
| ١ | ha eliminato: ) could be to assign |  |

ha formattato: Tipo di carattere: Corsivo

ha eliminato: methanol,

| ha eliminato: ;                           |  |  |  |
|-------------------------------------------|--|--|--|
| ha formattato: Tipo di carattere: Corsivo |  |  |  |
| ha eliminato: ).                          |  |  |  |
| ha eliminato: reluctance                  |  |  |  |
| ha eliminato: tier                        |  |  |  |
| ha formattato: Tipo di carattere: Corsivo |  |  |  |
| ha formattato: Tipo di carattere: Corsivo |  |  |  |
| ha formattato: Tipo di carattere: Corsivo |  |  |  |
| ha formattato: Tipo di carattere: Corsivo |  |  |  |
| ha eliminato: u <sub>emi</sub>            |  |  |  |

| 689 | $u_{AD} = \pm 5\%$ suggested by <i>IPCC-06</i> , while for Tier 1 the suggested $u_{EF}$ is of $\pm 60\%$ . $u_{EF}$ for other |
|-----|--------------------------------------------------------------------------------------------------------------------------------|
| 690 | carbonates (e.g. limestone) is due to the variability in composition and is very low ( $\sim 1\frac{6}{2}$ to 5%), while       |
| 691 | $u_{AD}$ can be much larger due to poor quality statistics and is assumed of $\pm 35\%$ .                                      |

Production of ammonia, nitric and adipic acid as well as caprolactam, glyoxylic and glyoxylic acid is 693 known with high degree of accuracy and  $u_{AD}$  for these processes can be estimated as  $\pm 2\%$ . The corresponding u<sub>EF</sub> is reported in Table 2\_Table and Table 3 and is derived from expert judgment 694 elicitation and reported in *IPCC-06* ( $\mu_{FF}^{Ammonia} = \pm 7\%$ ;  $u_{FF}^{Nitric Acid} = \pm 20\%$ ;  $u_{FF}^{Carbide} = \pm 10\%$ ). For 695 petrochemical and carbon black production (methanol, ethylene, ethylene dichloride, vinyl, 696 697 acrylonitrile, carbon black), IPCC-06 provides reference values for uEMI associated to these processes 698 (IPCC-06, Volume 3, Chapter 3, Table 3.27), based on expert judgments. The values are reported in 699 Table 3, ranging from  $\pm 10\%$  for CH<sub>4</sub> emission for ethylene production to  $\pm 85\%$  for CH<sub>4</sub> emission from 700 carbon black production.

As summarised in Table 1, the AD for iron and steel (including furnace technologies) production are considered very accurate, with  $u_{AD} = \pm 10\%$ , and for ferroalloys  $u_{AD}$  is set to  $\pm 10\%$  for industrialised countries and  $u_{AD} = \pm 20\%$  for developing countries, based on own judgment (*IPCC-06* suggests  $u_{AD} =$ 

±5%). The data for iron production are updated monthly using data from the World Steel Association

(WSA, 2019), while for ferroalloys data are extrapolated using trends from USGS commodity statistics

(USGS, 2016).  $u_{EF}$  is equal to  $\pm 25\%$ .

Production data for aluminium, magnesium, zinc, and lead are deemed accurate within 2% to 10%(Table 1). For aluminium, the reactions leading to CO<sub>2</sub> emissions are well understood and the emissions

are very directly connected to the quantity of aluminium produced (*IPCC-06*), and  $u_{\rm EF}$  is assumed within

10%. The u<sub>EF</sub> associated with CO<sub>2</sub> emitted from magnesium production is also well understood and is

711 assumed within 5%. Lead and zinc production have higher  $u_{EF}$  (50%) associated with default emission

factors (Tier 1), and of 15% if country specific data are adopted (Tier 2). CO<sub>2</sub> emissions for non-energy

vue of lubricants/waxes (like petroleum jelly, paraffin waxes and other waxes, classified under IPCC

sector 2.D.2 and corresponding to EDGAR sector NEU) are assumed highly uncertain (u<sub>EF</sub> of 100%;
 u<sub>AD</sub> of 5% to 15%) due to the lack of accurate information and to country specific operating conditions.

# 716 FIGURE 6.

# 717 CO<sub>2</sub> emissions in sector 2 are one and two orders of magnitude higher than N<sub>2</sub>O and CH<sub>4</sub> emissions

respectively (Figure 6). Nearly 50% of CO<sub>2</sub> emissions in this sector originate from cement production.

The accuracy ranges from medium-high to high for all top emitters, and the global uncertainty is of

12%. For  $N_2O$ , the main source (~85%) is the production of nitric and adipic acid, which results in

medium-high accuracy both country wise and globally. Finally, <u>emission of</u> CH<sub>4</sub> is more uncertain due

to the large  $u_{EF}$  of carbon black and methanol production, which account for ~52% of global CH<sub>4</sub> emissions in the IPPU sector.

# 724 3.1.4 AGRICULTURE

Agriculture related activities in EDGAR cover partially the IPCC category 3 (agriculture, forestry and

land use), including enteric fermentation (ENF, corresponding to 3.A.1), manure management (MNM,

3.A.2), waste burning of agricultural residues (AWB.CRP, corresponding to 3.C.1.b – biomass burning

of cropland), direct N<sub>2</sub>O emissions from soil due to natural and synthetic fertiliser use (corresponding

to 3.C.4), indirect N<sub>2</sub>O emissions from manure and soils (corresponding to 3.C.5 and 3.C.6), urea and  $\frac{1}{2}$ 

agricultural lime (AGS.LMN and AGS.URE, corresponding to IPCC codes 3.C.2 and 3.C.3), and rice

ha formattato: Tipo di carattere: Corsivo

# ha eliminato: ha eliminato: $u_{EF}^{ammonia} =$ ha formattato: Tipo di carattere: Corsivo ha formattato: Tipo di carattere: Corsivo ha formattato: Tipo di carattere: Corsivo

ha formattato: Tipo di carattere: Corsivo

ha formattato: Tipo di carattere: Corsivo

ha eliminato:

ha eliminato:

ha eliminato: Agriculture

cultivation (AGS.RIC corresponding to 3.C.7). Forestry and land use are not covered. <u>Data sources for</u>
 AD covering the agriculture sector are compiled by *Janssens-Maenhout et al.* (2019).

For sectors ENF and MNM, EDGAR follows IPCC-06 for estimating emissions, with animal counting 738 739 data from FAOSTAT (2018). For ENF, uncertainty in AD is due to cattle numbers, feed intake, and feed composition, while for MNM the distribution of manure (volatile solids) in different manure 740 management systems is also a source of uncertainty.  $u_{AD}$  for these sectors is estimated of ~±20% to 741 742 account for uncertainty of the manure management system usage, lack of detailed characteristics of 743 livestock industry, information on how manure management is collected, and lack of homogeneity in 744 the animal counting systems (IPCC-06; Olivier, 2002). The estimate is slightly higher than u<sub>AD</sub> from 745 other USA studies for ENF (EPA, 2017; Hristov et al., 2017), whilst for MNM uAD of ±20% might be 746 underestimated according to e.g. Hristov et al. (2017). EFs are calculated following IPCC\_06 747 methodology, using country specific data of milk yield and carcass weight integrated with trends from 748 FAOSTAT (2018) for cattle, and using regional EFs for livestock. Tier 1 uEF for ENF and MNM is 749 estimated to be larger than  $\pm 50\%$  (with a minimum of 30%) unless livestock characterisation is known 750 with great accuracy, in which case Tier 2 uncertainty can be  $\sim \pm 20\%$  (IPCC-06).

AD for burning of agriculture waste (AWB.CRP) can be highly uncertain, especially in developing 752 countries, due to several factors including the estimates of the area planted under each crop type for 753 which residues are normally burnt and the fraction of the agricultural residue that is burnt in the field. 754 EDGAR estimates the fraction of crop residues removed and/or burned using data from Yevich and 755 Logan (2003) and from official country reporting. Uncertainty is deemed very high, in the range 756  $u_{AD}^{AWB.CRP} \approx 50 \text{ to } 100\%$  (Olivier, 2002; Olivier et al., <u>1999a)</u>. EFs for this sector are obtained from 757 the mass of fuel combusted, provided by IPCC-06 as default (Tier 1) EFs for stationary combustion in the agricultural categories, and are estimated with an uncertainty of  $\sim$ -60% to +275% for N<sub>2</sub>O, and 758 759  $\sim \pm 50\%$  to  $\pm 150\%$  for CH<sub>4</sub>, according to the uncertainty for combustion processes.

Emissions from rice cultivation are relevant to CH<sub>4</sub>. According to the last release of EDGAR, in 2015 760 761 almost 10% of total CH4 emissions were due to rice cultivation. Default, baseline EF for rice cultivation 762 has an uncertainty in the range -40% to +70%, which has been substantially reviewed in the IPCC refinement (2019), both in terms of EF value and of uncertainty. The refinement also gives regional-763 764 dependent EF and uncertainty ranges, but those have not been implanted yet in EDGAR, therefore we 765 refer to the IPCC-06 guidelines. In EDGAR the baseline EF is multiplied by a set of scaling factors that 766 account for the water regimes before and during the cultivation period: upland (UPL, never irrigated), irrigated (IRR), rain fed (RNF) and deep water (DWP), which are assigned the following uncertainty 767 768 (derived from  $IPCC_{\underline{-06}}$ ): IRR  $\underline{=}$  -20% to +26%; UPL  $\underline{=}$  0%; RNF and  $\underline{DWP}$  = -22% to +26%. Organic 769 amendments and soil type are not included. The AD consist of cultivation period and annual harvested 770 area for each water regime and are derived from FAO (2011) and are complemented with data from 771 IRRI (2007) and IIASA (2007). We assume uAD of 5% to 10% (Olivier, 2002). All the conditions together 772 yield an uncertainty range of -0.45% to +75% for RNF, DWP and IRR, and of -0.41% to +70% for 773 UPL.

#### 774 **FIGURE 7**.

AD for sectors 3.C.2 (CO<sub>2</sub> emissions from liming), 3.C.3 (CO<sub>2</sub> emissions from urea application), are derived from *FAOSTAT* (2016), and from official country reporting. Uncertainty of emissions of CO<sub>2</sub> from lime (urea) fertilization stems from uncertainties in the amount of urea applied to soils and from the uncertainties in the quantity of carbonate applications that is emitted as CO<sub>2</sub>.  $u_{AD}$  is assumed of 20%

(*Olivier et al., <u>1999a</u>*) to account for uncertainty in sales, import/export and usage data adopted to derive

ha eliminato: Derivation of ...ata sources for AD for...overing the agriculture sector are compiled by *Janssens-Maenhout et al.*,

ha eliminato: ...f ...he ...anure ...anagement ...ystem usage, lack of detailed characteristics ...f each country's livestock ...ndustry and how ... information ...n how manure ...anagement ...s ...collected, and lack of homogeneity in the animal counting systems

# ha eliminato: et al., ha formattato

#### ha formattato: Tipo di carattere: Corsivo

ha eliminato: US...SA studies for ENF (*EPA*, 2017,... *Hristov et al.*, 2017), whilst for MNM u<sub>AD</sub> of ±20% might be underestimated according to, e.g.

**ha eliminato:** (2006)...06 methodology, using country specific data of milk yield and carcass weight integrated with trends from *FAOSTAT* (2018) for cattle, and using regional EFs for livestock. Tier 1 u<sub>EF</sub> for ENF and MNM is estimated to be larger than  $\pm$ 50%,

#### ha formattato

ha eliminato: ...lanted ...nder ...ach ...rop ...ype ...or which ...esidues ...re ...ormally

ha eliminato: toto 100% (...livier and Peters... 2002; Olivier et al., 1999). ...999a). EFs for this sector are obtained from the ...ass ...f ...uel

ha formattato: Tipo di carattere: Corsivo

**ha eliminato:** ... according to the uncertainty for combustion processes. ¶

|   | <b>ha eliminato:</b> , 200606): IRR:= -20% to +26%;<br>UPL:= 0%; RNF and DWE: |  |
|---|-------------------------------------------------------------------------------|--|
| - | ha formattato: Tipo di carattere: Corsivo                                     |  |
|   |                                                                               |  |

ha eliminato: DWE

#### ha eliminato: ¶

# ha formattato: Tipo di carattere: Corsivo

**ha eliminato:** 1998...999a) to account for uncertainty in sales, import

| 862<br>863 | the AD. EFs are derived from <i>IPCC-06</i> Tier 1, assuming that all C in urea is lost as CO <sub>2</sub> in the atmosphere, which might give rise to systematic bias. $u_{EF}$ is assumed between ±50% and ±100%. |                   | ha formattato: Tipo di carattere: Corsivo<br>ha eliminato: ranging |
|------------|---------------------------------------------------------------------------------------------------------------------------------------------------------------------------------------------------------------------|-------------------|--------------------------------------------------------------------|
| 864        | Sectors 3.C.4, 3.C.5, 3.C.6 cover direct and indirect N <sub>2</sub> O emissions from managed soils and manure                                                                                                      |                   | ha eliminato: ¶                                                    |
| 865        | management. AD are taken from FAOSTAT (2016) and UNFCCC (2018). Nitrogen from livestock data                                                                                                                        |                   | ha eliminato: data                                                 |
| 866        | for developed countries is derived from the CAPRI model (Leip et al., 2011) and can be considered as                                                                                                                | $\searrow$        | ha eliminato: ).                                                   |
| 867        | Tier 3 level accuracy. Indirect N <sub>2</sub> O emissions are due to leaching and runoff of nitrate and are subject                                                                                                |                   |                                                                    |
| 868        | to various sources of uncertainty (both AD and EFs) due to natural variability and to the volatilization                                                                                                            |                   |                                                                    |
| 869        | and leaching factors, poor measurement coverage and under-sampling as well as due to                                                                                                                                |                   |                                                                    |
| 870        | incomplete/inaccurate/missing information on observance of laws and regulations related to handling                                                                                                                 |                   |                                                                    |
| 871        | and application of fertiliser and manure, and changing management practices in farming ( <i>IPCC-06</i> ).                                                                                                          | _                 | ha formattato: Tipo di carattere: Corsivo                          |
| 872        | For these sectors, $u_{AD}$ is estimated $\pm 20\%$ and $u_{EF}$ in the range $\pm 65\%$ to $\pm 200\%$ according to <i>[PCC-06</i> ).                                                                              |                   | ha formattato: Tipo di carattere: Corsivo                          |
| 873<br>874 | Studies by, e.g., <i>Philibert et al.</i> (2012) and <i>Berdanier and Conant</i> (2012) suggest that the uncertainty of N <sub>2</sub> O emissions due to N fertilization can be as lower as up to a factor 5.      |                   |                                                                    |
| 875        | The large variation of $N_2O$ emissions in time and space is well recognised (e.g. <i>Stehfest and Bouwman</i> ,                                                                                                    |                   | ha eliminato: ¶                                                    |
| 8/6        | 2000). Spatial neterogeneity, in particular, is largely driven by soil properties, and the influence of soil                                                                                                        |                   |                                                                    |
| 8//        | En (Milno et al. 2014)                                                                                                                                                                                              | _                 | ha aliminatas emissien fostera                                     |
| 0/0        | <u>pris</u> ( <i>Mune et u.</i> ., 2014).                                                                                                                                                                           |                   | na emmato: emission factors                                        |
| 879        | With a few exceptions, the confidence in emission estimates from agriculture varies between medium                                                                                                                  |                   | ha eliminato: ¶                                                    |
| 880        | and low for CO <sub>2</sub> and CH <sub>4</sub> (Figure 7a,b) depending on the composition of the agricultural sources and                                                                                          | Ń                 | ha eliminato: sector                                               |
| 881        | on the accuracy assigned to the specific country (developing vs industrialised). No C (Figure 7c)                                                                                                                   |                   | ha eliminato: N <sub>2</sub> O (Figure                             |
| 882<br>883 | emissions are very uncertain (in excess of 300%), which is reflected in the global share of uncertainty (over 90%, though the share of global N <sub>2</sub> O emissions does not exceed 30%, Figure 7d).           |                   | Codice campo modificato                                            |
| 1          |                                                                                                                                                                                                                     | l<br>l            |                                                                    |
| 884        | For the UK, <i>Milne et al.</i> (2014) estimated a 95% <u>confidence interval of -56% to +139%</u> , <i>Brown et al.</i>                                                                                            |                   | ha eliminato: CI                                                   |
| 885        | (2012) of $-93%$ to +253%, whereas Monni et al. $(2007)$ of $-52%$ to +70% for Finland (but based on                                                                                                                |                   | ha eliminato:                                                      |
| 886        | older and more conservative <u>IPCC</u> guidelines). Our uncertainty estimates for the UK for sectors 3.C.4,                                                                                                        |                   | ha eliminato:                                                      |
| 887        | 3.C.5, 3.C.6 combined is of -/4% to 305% (as <u>direct</u> effect of assuming full correlation <u>; in fact</u> if the                                                                                              | $\square$         | ha eliminato:                                                      |
| 888        | three sectors were considered to be uncorrelated, the 95% confidence interval for the OK would be -                                                                                                                 | $\square$         | ha eliminato: ,                                                    |
| 889        | $59\%$ to $\pm 259\%$ , which is in the with the other estimates).                                                                                                                                                  | $\mathbb{N}$      | ha eliminato: are                                                  |
| 890        | FIGURE 8.                                                                                                                                                                                                           | $\langle \rangle$ | ha eliminato: CI                                                   |
|            |                                                                                                                                                                                                                     | Y                 | ha eliminato: is                                                   |
| 891        | Uncertainties due to rice cultivation and enteric fermentation outweigh the uncertainty from other                                                                                                                  |                   |                                                                    |
| 892        | sources, being the dominant emission shares over the emissions from burning of crop residues (which                                                                                                                 |                   | ha eliminato: due to                                               |
| 893        | has higher uncertainty but low impact on overall emission) (Figure 8). Agricultural uncertainties in                                                                                                                |                   |                                                                    |
| 894        | China are attributable to rice cultivation for $\sim 80\%$ , whilst rice emission accounts for less than 60% of                                                                                                     |                   |                                                                    |
| 895        | agriculture total. Similarly, the uncertainty due to enteric termentation dominates the USA                                                                                                                         |                   |                                                                    |
| 890        | agriculture uncertainty (75% share).                                                                                                                                                                                |                   |                                                                    |
| 897        | 3.1.5 WASTE                                                                                                                                                                                                         | (                 | Formattato: SpazioPrima: 0 pt, Dopo: 10 pt                         |
| 898        | The waste-related emissions in EDGAR correspond to IPCC category 4 (waste), including emissions                                                                                                                     |                   | ha eliminato: Waste                                                |
| 899        | from managed and non-managed landfills (SWD: solid waste disposal on land and incineration,                                                                                                                         |                   |                                                                    |
| 900        | categories 4.A, 4.B and 4.C), wastewater handling (domestic WWT.DOM and industrial WWT.IND,                                                                                                                         |                   |                                                                    |
| 901        | categories 4.D.1 and 4.D.2, emitting CH <sub>4</sub> and N <sub>2</sub> O), and waste incineration (emitting CH <sub>4</sub> , N <sub>2</sub> O, and                                                                |                   |                                                                    |
| 902        | also CO <sub>2</sub> ). Globally, the waste sector accounts for 4.4% of total GHG anthropogenic emission in 2015                                                                                                    |                   | ha eliminato: N <sub>2</sub> O and                                 |
| 903        | and 21.5% of total anthropogenic CH <sub>4</sub> emissions ( <i>Crippa et al.</i> , 2019).                                                                                                                          |                   |                                                                    |
|            | 12                                                                                                                                                                                                                  |                   |                                                                    |
|            | 13                                                                                                                                                                                                                  |                   |                                                                    |
|            |                                                                                                                                                                                                                     |                   |                                                                    |

In EDGAR, emissions are based on a combination of population and solid and liquid waste product statistic. CH<sub>4</sub> emissions from landfills are calculated following the first order decay model proposed by 925 926 IPCC-06, which assumes that emissions do not occur instantaneously but are spread over several years. 927 The model depends on several parameters (Table 1 and Table 3), and the main factor in determining 928 the CH<sub>4</sub> generation potential is the amount of degradable organic carbon (DOC) (*IPCC-06*; *Olivier*, 929 2002; Janssens-Maenhout et alg. 2019). The average weight fraction of DOC under aerobic conditions is provided by the IPCC Waste Model for 19 regions, which has been used as the default for all 930 931 countries. Moreover, the default parameters for the methane correction factor (MCF), constant (k) and 932 the oxidation factor (OX) are adopted (full details in Table 1 of Janssens-Maenhout et al. (2019). Each 933 component of the waste model has been assigned a normal distribution using the 95% confidence 934 interval defined in Table 1 and Table 3 and combined using a sample population of 10000 elements. The range of overall uncertainty is between 35% and 134% for CH<sub>4</sub> and between 10% and 490% for 935

N<sub>2</sub>O.

# For the incineration of waste, AD are derived from UNFCCC NIR, IPCC-06, country reports and scientific literature, extrapolated using population trends (e.g. for countries with scarce data on municipal solid waste), while for composting (category 'other'), data are obtained from UNFCCC NIR for Annex I countries and scientific literature for developing countries and for India (Table 1 of Janssens-Maenhout et al. (2019) and references therein).

As detailed in Janssens-Maenhout et al. (2019), the JPCC-06 default values for wastewater generation 943 and chemical oxygen demand (COD) are used to derive the total organically degradable material 944 (TOW), differentiating by type of industry (meat, sugar, pulp, organic chemicals, ethyl alcohol). 945 Population from UNHABITAT statistics (UNHABITAT, 2016) is used to derive country-specific 946 percentages of population at mid-year residing in urban and rural areas, with low and high income, for 947 calculating domestic wastewater. Different wastewater treatments are specified with technologyspecific CH4 emission factors. For domestic wastewater, the sewer to wastewater treatment plants 948 (WWTP), sewer to raw discharge, bucket latrine, improved latrine, public or open pit and septic tank 949 950 are distinguished. Uncertainty of domestic wastewater depends on the technology (sewer to raw 951 discharge, bucket latrine, improved latrine) as specified in Table 1 and Table 3, and is composed of 952 uncertainty in AD (population data <u>~±</u>36%) and uncertainty on EF (-33% to 78%).

Uncertainty on AD for industrial wastewater data ranges between -56% to 103%, estimated using the 954 *IPCC-06* suggested values, which are in line with those provided by *Olivier et al.* (2002) (-50% to 955 100%). Uncertainty on EF includes 30% uncertainty for the maximum CH<sub>4</sub> producing capacity 956 (parameter B<sub>0</sub>) and uncertainty on the CH<sub>4</sub> correction fraction of <u>-</u>50% to 100% (based on the range of 957 default values for MCF provided by *IPCC-06* in table 6.8 of Volume 5).

958Emissions of CH4 from the waste sector is one order of magnitude higher than  $N_2O$  and two orders959higher than  $CO_2$  (Figure 9a,b,c) and although  $N_2O$  emissions are more uncertain, the share of uncertainty960still reflects the share of emissions (Figure d). The confidence in the emission estimates varies from961medium to medium low for  $CO_2$  (depending on the status of development of the country), from medium962to very low for CH4 (depending on the status of development of the country and on the composition of963the waste sector, discussed below) and is very low for  $N_2O$  (due to high  $u_{EF}$  in waste water).

# 964 FIGURE 9.

965The composition of the waste sector for CH4 (Figure 10Figure ) shows that there is a strong966correspondence between the emissions share and the uncertainty share. For the USA, landfills emissions

#### ha formattato: Tipo di carattere: Corsivo

# ha eliminato: et al.,

|   | ha formattato: Tipo di carattere: Corsivo |
|---|-------------------------------------------|
| ľ | ha formattato: Tipo di carattere: Corsivo |
| ľ | ha formattato: Tipo di carattere: Corsivo |
| Y | ha eliminato: .                           |
| 1 | ha eliminato: ).                          |
| ۲ | ha eliminato: CI                          |
| 1 | ha eliminato:                             |
|   |                                           |
| - | ha eliminato: and                         |
|   |                                           |
| - | ha eliminato: ) ,                         |
| 1 | ha formattato: Tipo di carattere: Corsivo |
| - | ha eliminato:                             |

| ha formattato: Tipo di carattere: Corsivo |
|-------------------------------------------|
| ha eliminato: CODs                        |
| ha eliminato: TOWs                        |
| ha formattato: Tipo di carattere: Corsivo |

| ha elim | inato: (                           |
|---------|------------------------------------|
| ha form | attato: Tipo di carattere: Corsivo |
| ha form | attato: Citazione Carattere        |
| ha forn | attato: Citazione Carattere        |
| ha form | attato: Tipo di carattere: Corsivo |

ha eliminato: next

ha eliminato: Figure 10)

account for ~73% of waste emissions, and the uncertainty due to landfills is ~90%. In India, domestic
wastewater accounts for over 85% of waste emissions, driving the overall uncertainty with 97%.

### 984 FIGURE 10.

- Worldwide, the CH<sub>4</sub> emission share from landfills and domestic wastewater is approximately equivalent
- (~44% and ~41%, respectively), whilst landfills have a relatively larger weight in the global uncertainty
- share (~55% and ~41%, respectively).

#### 988 3.2 THE GLOBAL AND EUROPEAN PICTURE

- The values in Table 6 summarise the global uncertainty ranges. First the uncertainties are given for each
- sector and gas individually, then for the sum of the three GHGs for each sector, and finally for the sum
- of the three GHGs and for all the sectors together. The last row of the table, thus, is the overall EDGAR
- uncertainty on the worldwide GHG emissions.

#### 993 TABLE 6.

Globally, while CO<sub>2</sub> is by far the largest emitted GHG (in excess of 75%) followed by CH<sub>4</sub> (19%), the main source of uncertainty (~50%) is N<sub>2</sub>O (Figure 11a), followed by CH<sub>4</sub> (~29%). Agriculture alone accounts for 39% of the global uncertainty (Figure 11b) and is almost entirely due to N<sub>2</sub>O as discussed earlier (Figure 8d) and energy accounts for 44% (almost half of the uncertainty for energy is due to

998 N<sub>2</sub>O, Figure 1f) and waste (11%, driven by CH<sub>4</sub> emissions, Figure 9d).

# 999 FIGURE 11.

The picture is quite similar for EU27+UK (Figure 12) with the main difference being the larger 1001 uncertainty share of  $N_2O$  (~70%) due to the higher level of accuracy associated with CO<sub>2</sub> and CH<sub>4</sub>.

#### 1002 FIGURE 12.

4 UNCERTAINTY DUE TO METHODOLOGY

The considerable number of 'degrees of freedom' influencing the uncertainty of an emission inventory 1004 1005 such as EDGAR is *itself* a source of uncertainty originating from different methodological assumptions. 1006 As such, the structural uncertainty of emissions tackled in the previous section is subject to variability due to the sets of assumptions, methods, choices adopted for its quantification. It originates from lack 1007 of agreement/incomplete knowledge on the processes governing the emission sources and their 1008 1009 representativeness. Such a methodological uncertainty reflects the judgment of the uncertainty emission compiler and can give rise to a significant share of the overall uncertainty estimate. For instance, two 1010 1011 experts could suggest two different probabilistic models for the value of a certain emitting source, 1012 leading to a certain degree of variability in the PDFs of that source. Methodological uncertainty, thus, 1013 may arise from the assumptions adopted assessment, particularly when there are no clear guidelines or 1014 reference cases about methodological choices that allow comparability between evaluations.

One of the most impactful assumptions of this study is the correlation between subcategories/fuels and,
for the same category/fuel, between countries. This has a profound impact on the uncertainty estimate,
for example in inter-comparison studies where EDGAR's uncertainties are shown next to other
inventories whose uncertainty estimates do not account for correlation (e.g. *Petrescu et al., 2020*; *Choulga et al., 2020*).

FIGURE 13.

ha eliminato: accounts

| ha formattato: Evidenziato                               |  |
|----------------------------------------------------------|--|
| ha formattato: Evidenziato                               |  |
| ha eliminato: by gas and categories, including           |  |
| ha formattato: Evidenziato<br>ha formattato: Evidenziato |  |
|                                                          |  |
| ha eliminato: 7                                          |  |
| ha eliminato: gas                                        |  |
| ha eliminato: Figure                                     |  |
| ha formattato: Pedice                                    |  |
| ha eliminato: Figure b) and                              |  |

ha eliminato: ¶

| ha | eliminato: the    |
|----|-------------------|
| ha | eliminato: arises |

The global weight of the correlation is reflected in the total of Figure 13, where the uncertainty ranges

from 4% (no correlation) to above 20% for the correlated cases. The impact of assuming correlation of

1033 the uncertainties when aggregating the emissions of several countries outweighs any other assumptions.

For instance, the assumption to constrain the N<sub>2</sub>O uncertainty for energy in the range  $\pm 50\%$  to  $\pm 150\%$ has, globally, much lower impact over the total uncertainty (23% rather than 20%).

# 1036 FIGURE 14.

As shown in Figure 14 for EU27+UK, the effect of correlation on the variability of the uncertainty is considerable. Emissions from the energy sector are estimated <u>to be</u> accurate, <u>since</u> the 95% <u>confidence</u> interval lies within 2% of mean value when no correlation is assumed across countries, and <u>within</u> 7% when the correlation is set to one. The uncertainty of 13% for the Tier1 'default case' reflects the high share of uncertainty due to N<sub>2</sub>O since the only difference between the 'T1 default' and 'T1+OJ N<sub>2</sub>O' for energy is the upper limit of N<sub>2</sub>O uncertainty to ±50% and ±<u>150%</u> (OJ: Own Judgment). The same argument applies to the other sectors, most notably to agriculture (130% vs 36%, with or without

correlation), and is reflected in the total GHG emissions (15% vs 4%).

<u>Important to notice</u> that if EU27+UK <u>report</u> emissions as a single party, even Tier 1 propagation 1046 methods would return an accuracy comparable to the combination of independent estimates (i.e. as if 1047 all EU parties used independent, Tier 2 or 3 estimates of their emissions).

The comparison between the 'default' uncertainty ranges and 'EDGAR in-house expert judgment' for 1049 N<sub>2</sub>O shows the impact of choices on the quantification of the uncertainty, contributing to enhance the 1050 uncertainty variability. The case of energy in Figure 14 is an example: the default uncertainty of 13% 1051 can vary as much as 46% (down to 7%) due to different judgments in estimating  $u_{EF}$ .

# 1052 5, CONCLUSIONS

This study quantifies the structural uncertainty of the EDGAR inventory of GHGs. Given the widespread applications of EDGAR in many areas – modelling, policy, evaluation, planning – the qualification of its accuracy and quantification of its uncertainty are essential added values.

EDGAR is a consistent database based, predominantly, on Tier 1 methods to quantify emission from anthropogenic sources (on a three-level of sophistication, Tier 1 is the simplest). As such, the uncertainty analysis presented here follows the corresponding Tier 1 approach for uncertainties, also suggested by *IPCC (2006; 2019)* to assist in country reporting. Some additional assumptions have been put forward to allow for the simple Tier 1 uncertainty method to integrate with the EDGAR global database.

The global, comparable nature of EDGAR is one of its main attractiveness. Zooming in individual countries, the accuracy of EDGAR cannot, in general terms, match that of the country's inventory reporting panel who might adopt higher tiers for estimating emissions and uncertainties. Hence, it is when looking at cross-sector, gases and countries aggregation that the analysis presented in this study shows its benefits.

For the aggregation of emitting sources sharing the same underlying methodology, we have assumed that the uncertainty is amplified, and therefore the aggregation must account for their correlation. The correlation is kept <u>also</u> when aggregating the same sectors across countries and when aggregating subcategories, with some exceptions and caveats detailed in the main text.

To summarise:

ha formattato: Tipo di carattere: 11 pt

ha eliminato: 50% and  $\pm 100$ 

| - | ha formattato: Tipo di carattere: 11 pt |
|---|-----------------------------------------|
| - | ha eliminato: as                        |
| 1 | ha eliminato: as                        |
| ١ | ha eliminato: CI                        |
| ١ | ha eliminato: of                        |
| - | ha eliminato: 100.                      |
| - | ha eliminato: arguments apply           |
|   |                                         |

| ha eliminato: This simple reasoning suggests |
|----------------------------------------------|
| ha eliminato: reported its                   |
| ha eliminato:                                |

ha eliminato:

ha formattato: Tipo di carattere: Corsivo

ha eliminato: When

ha eliminato: are aggregated

ha formattato: Inglese (Regno Unito)

- <u>global</u> CO<sub>2</sub> emitted from the energy sector alone (IPCC sector 1) accounts for 96% of global
   GHG<u>emissions</u>, and is accurate within 7% (generally, high confidence levels for top emitters);
- when adding CH<sub>4</sub> and N<sub>2</sub>O, the accuracy of the energy sector decreases to an uncertainty of 12.8 to +15.9%;
- 089 the uncertainty of N<sub>2</sub>O for the power industry sector (factor of 10 suggested by  $IPCC_{-06}$ ) 090 indicates a very poor accuracy. This high value reflects the paucity of accurate estimates 091 although some studies suggest lower uncertainty values (Lee et al., 2013; Olivier et al., 1999 1092 For N<sub>2</sub>O in sector 1.A we set  $u_{EF} = \pm 50$  to  $\pm 150\%$  (industrialized and developing countries, 1093 respectively), to yield a global uncertainty of 112%; CH4 emitted by the oil and gas extraction 1094 facilities is highly uncertain although the guidelines provide detailed uncertainties for all stages 1095 (extraction, storage, distribution, transmission) and differentiated by the level of development 1096 of the country. Due to the discrepancies with scientific literature and the number of parameters 1097 and components of this sector we have tested a more conservative estimate of  $u_{AD} = \pm 5$  and 1098  $\pm 15\%$  (industrialised and developing countries, respectively) and  $u_{EF} = \pm 100\%$  to all countries 1099 (u<sub>EF</sub> of 50% for country specific EF) when considering aggregation of sectors/countries which 1100 yield a global CH<sub>4</sub> uncertainty of -55%; +93%;
- <u>agriculture</u> emissions are dominated by CH<sub>4</sub> and N<sub>2</sub>O, with the uncertainty of the latter (over 300% on a global average) outweighing that of CH<sub>4</sub> due to large uncertainty in <u>EFs. At the</u> global scale, CH<sub>4</sub> uncertainty is driven by rice cultivation and enteric fermentation;
- waste is also a sector dominated by CH<sub>4</sub> emissions, followed by N<sub>2</sub>O. The uncertainty of the latter are very high (often exceeding 400%), while for CH<sub>4</sub> emissions, the share from landfills and domestic wastewater is approximately equivalent (~44% and ~41%, respectively), whilst landfills have a relatively larger weight in the global uncertainty share (~55% and ~41%, respectively).
- The strongest assumption, <u>made also</u> in previous studies, is the full correlation of subcategories and countries which introduces a further source of uncertainty – methodological uncertainty – that is very impactful. Uncertainty around methodological choices arises when there are different views about what constitutes the <u>"correct"</u> approach for optimum decision making. This form of uncertainty might be dealt with by agreeing on a <u>"reference case"</u> or on a list of methodological choices to allow comparability between different inventories.

The choice of methods can have a profound impact on the overall uncertainty assessment and needs to 1116 be <u>taken</u> into consideration when comparing inventories. For EU27+UK, for example, the choice to

- assume or not correlation among countries can result in a ~4-fold variability of the uncertainty (4% vs
  15%).
- ACKNOWLEDGEMENTS

The authors are thankful to Dr P.Bergamaschi (JRC) for his insightful suggestions and critical review.

FINANCIAL SUPPORT

This research has been supported by the European Commission, Horizon 2020 Framework Programme

(VERIFY, grant no. 776810). Margarita Choulga was funded by the CO2 Human Emissions (CHE)

project which received funding from the European Union's Horizon 2020 research and innovation

- programme under grant agreement no. 776186.
- DATA AVAILABILITY

| ha eliminato: Global                                                                                                                    |
|-----------------------------------------------------------------------------------------------------------------------------------------|
| ha eliminato: emission                                                                                                                  |
| ha eliminato: When                                                                                                                      |
| ha formattato: Pedice                                                                                                                   |
| ha eliminato: ;                                                                                                                         |
| ha eliminato: The                                                                                                                       |
| ha eliminato: guidelines                                                                                                                |
| ha formattato: Tipo di carattere: Corsivo                                                                                               |
| ha eliminato: The                                                                                                                       |
| ha formattato: Evidenziato                                                                                                              |
| <b>ha eliminato:</b> is however too high to be used in a comparative analysis and aggregation, as it masks the results and contribution |
| ha formattato: Evidenziato                                                                                                              |
| <b>ha eliminato:</b> other sources/gas. Therefore, expert judgement for                                                                 |
| ha formattato: Evidenziato                                                                                                              |
| ha eliminato: is to use                                                                                                                 |
| ha formattato: Evidenziato                                                                                                              |
| ha eliminato:                                                                                                                           |
| ha formattato: Evidenziato                                                                                                              |
| ha eliminato: ¶                                                                                                                         |
| ha eliminato: suggest to apply                                                                                                          |
| <b>ha eliminato:</b> produce uncertainty ranges more in line with the scientific literature to                                          |
| ha eliminato: Agriculture                                                                                                               |
| ha eliminato: emission factors.                                                                                                         |
| ha eliminato: Waste                                                                                                                     |
| ha eliminato: already used                                                                                                              |
| ha eliminato: "correct"                                                                                                                 |
| ha eliminato: "                                                                                                                         |
| ha eliminato: case''                                                                                                                    |
| ha eliminato: take                                                                                                                      |

ha eliminato: )

| 1154                                                                 | The database underlying the analysis is EDGARv5.0, it's open access and available at                                                                                                                                                                                                                                                                                                                                                                                                                                                                                                                                                                                                                                                                                                                         | Formattato: Giustificato                                                                                                                                                                                                                                                                                                                                                        |
|----------------------------------------------------------------------|--------------------------------------------------------------------------------------------------------------------------------------------------------------------------------------------------------------------------------------------------------------------------------------------------------------------------------------------------------------------------------------------------------------------------------------------------------------------------------------------------------------------------------------------------------------------------------------------------------------------------------------------------------------------------------------------------------------------------------------------------------------------------------------------------------------|---------------------------------------------------------------------------------------------------------------------------------------------------------------------------------------------------------------------------------------------------------------------------------------------------------------------------------------------------------------------------------|
| 1155                                                                 | https://edgar.jrc.ec.europa.eu/overview.php?v=50_GHG, last access: 15 January 2021.                                                                                                                                                                                                                                                                                                                                                                                                                                                                                                                                                                                                                                                                                                                          | ha eliminato:                                                                                                                                                                                                                                                                                                                                                                   |
| 1156                                                                 | AUTHOR CONTRIBUTION                                                                                                                                                                                                                                                                                                                                                                                                                                                                                                                                                                                                                                                                                                                                                                                          | ha eliminato:<br>https://edgar.jrc.ec.europa.eu/overview.php?v=50_GHG                                                                                                                                                                                                                                                                                                           |
| 1157                                                                 | E.Solazzo: design of the study, analysis, writing; M.Crippa, D.Guizzardi, M.Muntean: emission                                                                                                                                                                                                                                                                                                                                                                                                                                                                                                                                                                                                                                                                                                                | ha eliminato:                                                                                                                                                                                                                                                                                                                                                                   |
| 1158                                                                 | database; M. Choulga: support in the uncertainty analysis of CO2; G. Janssens-Maenhout: emission                                                                                                                                                                                                                                                                                                                                                                                                                                                                                                                                                                                                                                                                                                             | ha eliminato: -                                                                                                                                                                                                                                                                                                                                                                 |
| 1159                                                                 | database and design of the study.                                                                                                                                                                                                                                                                                                                                                                                                                                                                                                                                                                                                                                                                                                                                                                            | ha formattato: Pedice                                                                                                                                                                                                                                                                                                                                                           |
| 1160                                                                 | COMPETING INTERESTS                                                                                                                                                                                                                                                                                                                                                                                                                                                                                                                                                                                                                                                                                                                                                                                          | ha eliminato:                                                                                                                                                                                                                                                                                                                                                                   |
| 1161                                                                 | The authors declare that they have no conflict of interest.                                                                                                                                                                                                                                                                                                                                                                                                                                                                                                                                                                                                                                                                                                                                                  | ha eliminato: No competing interests¶                                                                                                                                                                                                                                                                                                                                           |
| 1162                                                                 | REFERENCES                                                                                                                                                                                                                                                                                                                                                                                                                                                                                                                                                                                                                                                                                                                                                                                                   | ¶<br>¶                                                                                                                                                                                                                                                                                                                                                                          |
| 1163<br>1164<br>1165<br>1166<br>1167<br>1168<br>1169<br>1170<br>1171 | <ul> <li>Andreae, M. O.: Emission of trace gases and aerosols from biomass burning – an updated assessment,<br/>Atmos. Chem. Phys., 19, 8523–8546, https://doi.org/10.5194/acp-19-8523-2019, 2019.</li> <li>Andreae, M. O. and Merlet, P.: Emission of trace gases and aerosols from biomass burning, Global<br/>Biogeochem. Cy., 15, 955–966, 2001.</li> <li>Andres, R. J., Boden, T. A., and Highdon, D.: A new evaluation of the uncertainty associated with<br/>CDIAC estimates of fossil fuel carbon dioxide emission, Tellus B, 66, 1–15,<br/>https://doi.org/10.3402/tellusb.v66.23616, 2014.</li> <li>Bernadier, A.B., Conant, R.T., 2012. Regionally differentiated estimates of croplands N2O emission<br/>reduce uncertainty in global calculations. Global Change Biology 18: 928–935</li> </ul> | Alice E. Milne, Margaret J. Glendining, Pat Bellamy, Tom<br>Misselbrook, Sarah Gilhespy, Monica Rivas Casado, Adele<br>Hulin, Marcel van Oijen, Andrew P. Whitmore, Analysis ol<br>uncertainties in the estimates of nitrous oxide and methane<br>emissions in the UK's greenhouse gas inventory for<br>agriculture, Atmospheric Environment, Volume 82, 2014,<br>Pages 94-105¶ |
| 1172<br>1173<br>1174<br>1175<br>1176<br>1177<br>1178<br>1179         | Bergamaschi, P., A. Danila, R. F. Weiss, P. Ciais, R. L. Thompson, D. Brunner, I. Levin, Y. Meijer, F. Chevallier, G. Janssens-Maenhout, H. Bovensmann, D. Crisp, S. Basu, E. Dlugokencky, R. Engelen, C. Gerbig, D. Günther, S. Hammer, S. Henne, S. Houweling, U. Karstens, E. Kort, M. Maione, A. J. Manning, J. Miller, S. Montzka, S. Pandey, W. Peters, P. Peylin, B. Pinty, M. Ramonet, S. Reimann, T. Röckmann, M. Schmidt, M. Strogies, J. Sussams, O. Tarasova, J. van Aardenne, A. T. Vermeulen, F. Vogel, Atmospheric monitoring and inverse modelling for verification of greenhouse gas inventories, EUR 29276 EN, Publications Office of the European Union, Luxembourg, 2018, ISBN 978-92-79-88938-7, doi:10.2760/759928, JRC111789                                                          |                                                                                                                                                                                                                                                                                                                                                                                 |
| 1180<br>1181<br>1182<br>1183<br>1184<br>1185                         | Bergamaschi, P., Corazza, M., Karstens, U., Athanassiadou, M., Thompson, R. L., Pison, I., Manning,<br>A. J., Bousquet, P., Segers, A., Vermeulen, A. T., Janssens-Maenhout, G., Schmidt, M., Ramonet,<br>M., Meinhardt, F., Aalto, T., Haszpra, L., Moncrieff, J., Popa, M. E., Lowry, D., Steinbacher, M.,<br>Jordan, A., O'Doherty, S., Piacentino, S., and Dlugokencky, E.: Top-down estimates of European<br>CH <sub>4</sub> and N <sub>2</sub> O emissions based on four different inverse models, Atmos. Chem. Phys., 15, 715–<br>736, https://doi.org/10.5194/acp-15-715-2015, 2015.                                                                                                                                                                                                                 |                                                                                                                                                                                                                                                                                                                                                                                 |
| 1186<br>1187<br>1188                                                 | Bond, T. C., D. G. Streets, K. F. Yarber, S. M. Nelson, JH. Woo, and Z. Klimont (2004), A technology-<br>based global inventory of black and organic carbon emissions from combustion, J. Geophys. Res.,<br>109, D14203, doi:10.1029/2003JD003697.                                                                                                                                                                                                                                                                                                                                                                                                                                                                                                                                                           |                                                                                                                                                                                                                                                                                                                                                                                 |
| 1189<br>1190                                                         | Brown, J.R., Blankinship, J.C., Niboyet, A. <i>et al.</i> Effects of multiple global change treatments on soil N <sub>2</sub> O fluxes. <i>Biogeochemistry</i> <b>109</b> , 85–100 (2012). https://doi.org/10.1007/s10533-011-9655-2                                                                                                                                                                                                                                                                                                                                                                                                                                                                                                                                                                         |                                                                                                                                                                                                                                                                                                                                                                                 |
| 1191<br>1192                                                         | Choulga, M, and et al., 2020. Global anthropogenic CO2 emissions and uncertainties as prior for Earth system modelling and data assimilation. ESSD Discussion, <u>https://doi.org/10.5194/essd-2020-68</u>                                                                                                                                                                                                                                                                                                                                                                                                                                                                                                                                                                                                   |                                                                                                                                                                                                                                                                                                                                                                                 |

- CIA: Central Intelligence Agency, The World Fact Book, Washington DC, available at: 1193 http://www.cia.gov/library/publications/ the-world-factbook (last access: October 2020), 2016. 1194
- CCRI: China Cement Research Institute (www.ccement.com), 2019 (last access: June 2020) 1195
- Ciais, P., C. Sabine, G. Bala, L. Bopp, V. Brovkin, J. Canadell, A. Chhabra, R. DeFries, J. Galloway, M. Heimann, C. Jones, C. Le Quéré, R.B. Myneni, S. Piao and P. Thornton, 2013: Carbon and 1196 1197 1198 Other Biogeochemical Cycles. In: Cli-mate Change 2013: The Physical Science Basis.

| a eliminato:                                             |
|----------------------------------------------------------|
|                                                          |
| a eliminato: No competing interests                      |
| "                                                        |
|                                                          |
|                                                          |
| ha eliminato: ¶                                          |
| Alice E. Milne, Margaret J. Glendining, Pat Bellamy, Tom |
| Misselbrook, Sarah Gilhespy, Monica Rivas Casado, Adele  |
| Hulin, Marcel van Oijen, Andrew P. Whitmore, Analysis of |

- Contribution of Working Group I to the Fifth Assessment Report of the Intergovernmental Panel
  on Climate Change [Stocker, T.F., D. Qin, G.-K.