# Peer review of "UNCERTAINTIES IN THE EDGAR EMISSION INVENTORY OF GREENHOUSE GASES"

_Atmospheric Chemistry and Physics, 2020_

## Referee Comment (RC1) · Anonymous Referee #1 · 29 Nov 2020

General Comments

The article describes the methodology used to quantify the structural uncertainty of the EDGAR inventory of GHGs. Although the methodology isn't conceptual new (it largely follows the one suggested by the IPCC guidelines) I agree with the authors that the qualification of its accuracy and quantification of its uncertainty are essential added values for the variety of users of the database and as such justify a paper with a detailed description of the method constructed. Although the value of the study lies more in the precise description of its method and choices made than in the exact outcome of the numbers (which of course will change in the future when new insights will lead to different values for certain u_AD and u_EF). At the end of the paper a short but valuable discussion is included about the effect of several of their choices on the

outcome of the calculated uncertainties.

Specific Comments

I found the sentence in L36 and further rather difficult to read, maybe reword for more easy reading?

I recommend to include a sentence somewhere at the beginning of the methodology section mentioning the fact that the presented uncertainties only relate to sources included in the database and as such say nothing about the missing sources and thus overall uncertainty of the total human induced emissions.

What is meant with the sentence in L97-98, especially the part 'than for recent years'?

What is the use of L160-163? Why not just put a period before ',allowing' and remove the rest since they are just an explanation of the words?

I have several related questions over table 5: first of all what is the source of table 5? No justification is given in the text. If it is the status of the country in the convention I think the whole Table can be left out since using the color legend in the figures is enough information. If the authors decided themselves which country is industrialized or developing I would expect a discussion why China is developing while several Caucasian countries (e.g. Armenia) are defined as industrialized. Even then Table 5 could be shortened by mentioning industrialized countries only and state the 'non mentioned' countries to be 'developing'.

L479 Why are numbers used of -0.45% and 0.41%? Are those significant or should they be reported as 0%?

Technical Comments

In the text several times a reference is made to Olivier and Peters (2002), Olivier et al., (2002) and Olivier (2002); in the reference list, however, only Olivier (2002) is found. Are all these articles the same or should the reference list extended?

L29 remove 'and for'

L52 replace 'assumes' by 'should have'?

L128 while the larger values [are assigned] to: add words between brackets?

L182 'upper' should be replaced by 'lower'

L189 typo u_Emi -> u_EMI

L225 typo is propagate -> is propogated

L340 remove 'in'

L473 define UPL after 'upland'

L474 is it DWP or DWE?

L585 add 'is' 'between and-almost'?

L683 typo be take -> be taken
* * *

---

## Referee Comment (RC2) · Antoon Visschedijk (Referee) · 17 Dec 2020

This is a highly relevant article for anyone that uses the EDGAR emission estimates in scientific research. Any research that uses EDGAR emission data as input, should verify to what extend research findings and conclusions are influenced by taking the uncertainty of the EDGAR emission estimates into account. EDGAR's applications in research are numerous and there is a strong, widely shared need for a peer reviewed reliable and robust assessment of the uncertainty of the estimated emissions.

To assess the uncertainty of the EDGAR emission data, the authors basically estimate the individual uncertainty of all components of the emission calculations performed by EDGAR, and then combine and aggregate these uncertainties to come to a total uncer-

tainty. EDGAR emission estimates consist of many separate emission source contributions, of which the uncertainties vary widely between source groups, substances and regions. The article currently under review is an essential reference to those EDGAR users who want to understand how uncertainty has been quantified, down to the level of individual combinations of substance, source and region. And because the EDGAR database is extensive by design, the article is (and should be) relatively long, detailed and complete to serve this purpose.

The length and degree of detail may make an article like this somewhat harder to review though, and I have not checked each individual component of the uncertainty assessment. I have found the numerous assumptions that need to be made in an assessment like this to be all adequately referenced and, as far as possible based on peer reviewed research.

As mentioned this article will serve as an essential and comprehensive reference to EDGAR users and it is therefore important that it gets published. The largest scientific value of the article is not so much in all these details but rather in what they amount to when correctly interpreted and combined in a statistically sound manner. The total global uncertainty of the emissions of the three discussed greenhouse gases in EDGAR is indeed a new and valuable scientific insight that was previously not available to the scientific community. In addition the methodology described appears consistent and statistically sound, and I would consider the methodology a valuable tool to apply in other local, regional and global emission databases.

General recommendations:

The article would probably benefit from review by a native English speaker (which I am not) and perhaps a somewhat more liberal use of comma's in long sentences.

An important finding of this research is also that for several sources of methane and especially nitrous oxide, current literature estimates of the emission factor uncertainties were found to be very high, in fact so high that the authors chose to replace these

uncertainty estimates by their own assumptions. The article might benefit from a some-what more elaborate substantiation of why these new uncertainty ranges were chosen, and what consequences the new ranges have for the total uncertainty. Please be aware that these assumptions are somewhat of a weaker spot in the research.

A welcome addition to the summary may be a short qualitative explanation of what happens to the total uncertainty when the emission totals would be disaggregated to the level of regions and/or individual countries. Does the (relative) uncertainty of the regional totals remain the same because of methodological dependencies, or increase considerably. Also good to mention in the summary that uncertainty of the spatial distribution in EDGAR is outside the scope of the article and not considered.

Lastly, different methodological choices for estimating uncertainties may result in a different outcome. Would you say that you are discussing the 'uncertainty of the chosen method to quantify the uncertainty?

Specific comments:

Line 26: Do you mean 'global sum' instead of 'global average'?

Line 29: 'accounts' should be 'account'

Line 36: 'of 1/4' should be '1/4rd'

Line 36: The sentence starting with 'Significant efforts..' might be rephrased as it is difficult to understand the point that is being made.

Lines 42 and 43: Not clear to me what you mean by 'les controversial and more responsive sources'

Line 57: Not clear to me what is meant by 'Scientific/assessmen/impact purposes'

Line 72: 'relaying' should probably be 'relying'

Line 81: Perhaps replace 'goodness' by 'quality'

Line 99 – 101: I would say that the primary source of emission factor uncertainty is the degree of representativeness of the limited number of observations underlying the emission factor, for the activity that is addressed. So rather the representativeness of the emission factor instead of any measurement errors.

Line 653: Please use subscript for the 2 in N2O

Lines 655 to 665: If you decide to add more text to explain why and how you chose to replace some literature ranges for CH4 and N2O uncertainties by your own assumptions, it would be good to include a reference here to that text

Line 676: would you rather say 'made' instead of 'used' here?

Recommendations regarding specific tables/figures:

Regarding Table 1 it may be good to mention in the text discussing this table that in general, small uncertainties are normally distributed, and large uncertainties (e.g. 'a factor of 5') are often lognormally distributed (this is just something that you may consider mentioning)

I have a difficulty understanding and interpreting Table 7. Could you give some additional explanation in the text what exactly the numbers in Table 7 mean and represent?

Please use sufficiently large font size in all figures. I have difficulty reading them because they are so small. You may decide to leave out some labels to make more room for larger characters, and the figures easier on the eye.

The 'look' of the bar charts may be improved. The bars look very bulky and all very similar (e.g. Figure 2 etc.), only the colors vary. You might consider a different way of presenting the data, but this would be up to you.

Please make sure that the text explaining the figures and the captions (notably figures 13 and 14) is sufficient to easily understand what the figures show/represent and how to interpret them. It took me quite some time to 'sort of' understand them and this

should not be necessary in my opinion.
* * *

---

## Author Comment (AC1) · 25 Jan 2021

**Author Comments to the Anonymous Referee #1 and Referee #2**

Comments to the manuscript of Efisio Solazzo et al. "Uncertainties in the EDGAR emission inventory of greenhouse gases"

**Anonymous Reviewer #1 comments and Authors reply**

General Comments

The article describes the methodology used to quantify the structural uncertainty of the EDGAR inventory of GHGs. Although the methodology isn't conceptual new (it largely follows the one suggested by the IPCC guidelines) I agree with the authors that the qualification of its accuracy and quantification of its uncertainty are essential added values for the variety of users of the database and as such justify a paper with a detailed description of the method constructed. Although the value of the study lies more in the precise description of its method and choices made than in the exact outcome of the numbers (which of course will change in the future when new insights will lead to different values for certain u\_AD and u\_EF). At the end of the paper a short but valuable discussion is included about the effect of several of their choices on the outcome of the calculated uncertainties.

**REPLY.** We thank the reviewer for the constructive comment, which accurately reflects the messages this study wants to convey, namely i) the added value given by the uncertainty associated with emissions estimates and ii) the numbers presented are not the ground truth, the method and order of magnitude of uncertainty estimates are the main value.

**Specific Comments**

I found the sentence in L36 and further rather difficult to read, maybe reword for more easy reading? **REPLY.** This sentence has been shortened and rephrased to simplify the reading:

'Measures put in place to attenuate temperature rise and to mitigate climate dynamics long-term changes, have contributed to uphold the role of CH4 and N2O. Their high warming potential compared to CO2 and relatively shorter life-time (on average CH4 persists in the atmosphere for approximately a decade, N2O for over a century and CO2 for more than 1000 years (NCR, 2010; Ciais et al., 2013)) allow to act on shifting from energy-related CO2 to other, more rapidly responsive, emission sources (Janssens-Maenhout et al., 2019; United Nations Environment Programme, 2019). At the same time, while for fossil fuel CO2 emissions the uncertainty is relatively small and, overall, well defined, for CH4 and N2O the emission estimates are significantly more uncertain. In turn, emission reduction measures issued by national plans highly depend on the degree of uncertainty of sectors that are supposed to contribute to reach the designed reduction target'

I recommend to include a sentence somewhere at the beginning of the methodology section mentioning the fact that the presented uncertainties only relate to sources included in the database and as such say nothing about the missing sources and thus overall uncertainty of the total human induced emissions. **REPLY.** The following text has been added at the end of section 2:

'This study addresses the uncertainty of the anthropogenic sources covered by EDGAR, which might be not exhaustive. Therefore, nothing can be said about the uncertainty stemming from source categories not currently encompassed within the inventory (e.g., fugitive  $CO_2$  from low temperature oxidation of coal mines, fugitive  $CH_4$  from managed wetlands,  $N_2O$  from crab ponds as part of aquaculture). Uncertainty assessment of spatially distributed sources (emission gridmaps) is beyond the scopes of this study.'

What is meant with the sentence in L97-98, especially the part 'than for recent years'? **REPLY.** This sentence has been rephrased in the following way: '*Similarly, AD of countries with transitional economies are expectedly more accurate for recent years.*'

What is the use of L160-163? Why not just put a period before 'allowing' and remove the rest since they are just an explanation of the words?

**REPLY.** We thank the reviewer for the comment. We have removed the final part of the sentence *'comparability, consistency, and transparency, allowing'*, but kept the description as it serves the scopes of the paper.

I have several related questions over table 5: first of all what is the source of table 5? No justification is given in the text. If it is the status of the country in the convention I think the whole Table can be left out since using the color legend in the figures is enough information. If the authors decided themselves which country is industrialized or developing I would expect a discussion why China is developing while several Caucasian countries (e.g. Armenia) are defined as industrialized. Even then Table 5 could be shortened by mentioning industrialized countries only and state the 'non mentioned' countries to be 'developing'.

**REPLY.** We agree with the reviewer, Table 5 has been moved to the supplementary material. The table is extracted from Janssens-Maenhout et al., 2019, where all details and rationale are provided. The caption to the table now contains the reference.

L479 Why are numbers used of -0.45% and 0.41%? Are those significant or should they be reported as 0%?

**REPLY.** We kept two significant digits as for the other estimated uncertainty, although to the effect of calculation the difference with 0% is indeed negligible.

**Technical Comments**

In the text several times a reference is made to Olivier and Peters (2002), Olivier et al., (2002) and Olivier (2002); in the reference list, however, only Olivier (2002) is found. Are all these articles the same or should the reference list extended?

**REPLY.** We have added necessary records in the references list.

L29 remove 'and for'. **REPLY.** Done.

L52 replace 'assumes' by 'should have'? **REPLY.** Done.

L128 while the larger values [are assigned] to: add words between brackets? **REPLY.** Done.

L182 'upper' should be replaced by 'lower' **REPLY.** Done.

L189 typo u\_Emi -> u\_EMI **REPLY.** Done.

L225 typo is propagate -> is propagated **REPLY.** Done.

L340 remove 'in' **REPLY.** Done.

L473 define UPL after 'upland' **REPLY.** Done.

L474 is it DWP or DWE? **REPLY.** Done.

L585 add 'is' 'between and-almost'? **REPLY.** Done.

L683 typo be take -> be taken **REPLY.** Done.

**Dr A. Visschedijk (Reviewer #2) comments and Author reply**

This is a highly relevant article for anyone that uses the EDGAR emission estimates in scientific research. Any research that uses EDGAR emission data as input, should verify to what extend research findings and conclusions are influenced by taking the uncertainty of the EDGAR emission estimates into account. EDGAR's applications in research are numerous and there is a strong, widely shared need for a peer reviewed reliable and robust assessment of the uncertainty of the estimated emissions. To assess the uncertainty of the EDGAR emission data, the authors basically estimate the individual uncertainty of all components of the emission calculations performed by EDGAR, and then combine and aggregate these uncertainties to come to a total uncertainty. EDGAR emission estimates consist of many separate emission source contributions, of which the uncertainties vary widely between source groups, substances and regions. The article currently under review is an essential reference to those EDGAR users who want to understand how uncertainty has been quantified, down to the level of individual combinations of substance, source and region. And because the EDGAR database is extensive by design, the article is (and should be) relatively long, detailed and complete to serve this purpose. The length and degree of detail may make an article like this somewhat harder to review though, and I have not checked each individual component of the uncertainty assessment. I have found the numerous assumptions that need to be made in an assessment like this to be all adequately referenced and, as far as possible based on peer reviewed research. As mentioned, this article will serve as an essential and comprehensive reference to EDGAR users and it is therefore important that it gets published. The largest scientific value of the article is not so much in all these details but rather in what they amount to when correctly interpreted and combined in a statistically sound manner. The total global uncertainty of the emissions of the three discussed greenhouse gases in EDGAR is indeed a new and valuable scientific insight that was previously not available to the scientific community. In addition, the methodology described appears consistent and statistically sound, and I would consider the methodology a valuable tool to apply in other local, regional and global emission databases.

**REPLY.** We thank Dr A. Visschedijk for the supportive comment, especially in recognising the relevance of this study in light of the use of the EDGAR inventory. We also appreciate the emphasis given to the importance for this work to be published.

General recommendations:

The article would probably benefit from review by a native English speaker (which I am not) and perhaps a somewhat more liberal use of comma's in long sentences.

**REPLY.** We have carefully revised the manuscript to improve readability and grammar.

An important finding of this research is also that for several sources of methane and especially nitrous oxide, current literature estimates of the emission factor uncertainties were found to be very high, in fact so high that the authors chose to replace these uncertainty estimates by their own assumptions. The article might benefit from a somewhat more elaborate substantiation of why these new uncertainty ranges were chosen, and what consequences the new ranges have for the total uncertainty. Please be aware that these assumptions are somewhat of a weaker spot in the research.

**REPLY.** The reviewer is indeed right. We have included our own judgment in two instances: for fugitive  $CH_4$  emissions and for  $N_2O$  emissions. The two cases are different.

For fugitive emissions of CH4 we have substantiated our judgment based on several available studies (*EPA*, 2017; *Littlefield et al.*, 2017; *Saunois et al.*, 2020; *Peischl et al.*, 2015; *Janssens-Maenhout et al.*, 2019; *Turner et al*, 2015; *Olivier et al.*, 2002) which give a sample of the large variability of the many components concurring to the total emission in this sector. On the other hand, *IPCC* (2006) guidelines provide, for this sector, uncertainty detailed to each single process and even divided up for developing and developed countries. The net result of the uncertainty propagation is Figure 4, which depicts a very large country to country variability, which seems to contrast with the literature cited above. For this reason, we suggest to end user to gauge the alternative uncertainty ranges suggested by

*Olivier et al.* (2002), for which we also provide the plot of Figure 5. Indeed Figure 5 is presented for completeness, as for the final aggregation we retain the IPCC values. It is now explained in the revised text.

For N2O, the scientific literature is not very detailed (to our knowledge) and, as highlighted by the reviewer, our own judgment might be a weak point here. We have substantiated our judgment with more references to studies where IPCC uncertainties are found to be high (*Lee et al., 2013; Philibert et al., 2012; Berdanier and Conant, 2012*), although their representativeness of conditions, technologies, etc. is limited. We assumed that, lacking detailed studies, an uncertainty lying in the 'red zone' of the plots (denoted in the plots as 'very poor confidence', above 100%) is indicative of the poor confidence in the emission estimate for that process. From a qualitative point of view, the message doesn't change. Quantitatively, a lower uncertainty values of N2O emissions (anyway supported by *Olivier et al. (2002)*) allows a more insightful aggregation with the other gases. The message remains clear, though: N2O emissions are highly uncertain and the confidence on them is rather poor.

The consequences to the total uncertainty of the choice to cap the  $N_2O$  uncertainty are summarised in Figure 13. We have now added the explanation in the revised text.

A welcome addition to the summary may be a short qualitative explanation of what happens to the total uncertainty when the emission totals would be disaggregated to the level of regions and/or individual countries. Does the (relative) uncertainty of the regional totals remain the same because of methodological dependencies, or increase considerably.

**REPLY.** The question is not entirely clear to us. For each source category, we have produced a plot reporting the top-emitting countries for that category, followed by the world total. For example: Figure 1a reports  $CO_2$  emissions and uncertainties for sector 1.A, displaying the top 30 emitting countries. The labels at the top report the top range of the uncertainty for each country. The world total  $CO_2$  emission for the sector 1.A is then displayed on the right-hand side. We believe that the question raised by the reviewer can be addressed by consulting the panel plots we have produced for each gas and source category.

Also good to mention in the summary that uncertainty of the spatial distribution in EDGAR is outside the scope of the article and not considered.

**REPLY.** This has been mentioned at the end of section 2, also in reply to a comment from Reviewer#1.

Lastly, different methodological choices for estimating uncertainties may result in a different outcome. Would you say that you are discussing the 'uncertainty of the chosen method to quantify the uncertainty?

**REPLY.** Yes, exactly. This part of the methodological uncertainty was initially much longer, but it was hard to explain, and was distracting from the main message of the article, that remains the structural uncertainty related to EDGAR. Nonetheless, from a quantitative perspective, a common ground for the aggregation of uncertainty is missing and comparability across studies is problematic.

Specific comments: Line 26: Do you mean 'global sum' instead of 'global average'? **REPLY.** Corrected.

Line 29: 'accounts' should be 'account' **REPLY.** Corrected.

Line 36: 'of 1/4' should be '1/4rd' **REPLY.** Addressed.

Line 36: The sentence starting with 'Significant efforts..' might be rephrased as it is difficult to understand the point that is being made. **REPLY.** Revised.

Lines 42 and 43: Not clear to me what you mean by 'les controversial and more responsive sources'

**REPLY. Revised.**

Line 57: Not clear to me what is meant by 'Scientific/assessment/impact purposes' **REPLY.** Revised.

Line 72: 'relaying' should probably be 'relying' **REPLY.** Corrected.

Line 81: Perhaps replace 'goodness' by 'quality' **REPLY.** Corrected.

Line 99 - 101: I would say that the primary source of emission factor uncertainty is the degree of representativeness of the limited number of observations underlying the emission factor, for the activity that is addressed. So rather the representativeness of the emission factor instead of any measurement errors.

**REPLY.** Revised.

Line 653: Please use subscript for the 2 in N2O **REPLY.** Corrected.

Lines 655 to 665: If you decide to add more text to explain why and how you chose to replace some literature ranges for CH4 and N2O uncertainties by your own assumptions, it would be good to include a reference here to that text.

**REPLY.** We have improved the description as suggested.

Line 676: would you rather say 'made' instead of 'used' here? **REPLY.** Corrected.

**Recommendations regarding specific tables/figures:**

Regarding Table 1 it may be good to mention in the text discussing this table that in general, small uncertainties are normally distributed, and large uncertainties (e.g. 'a factor of 5') are often lognormally distributed (this is just something that you may consider mentioning)

**REPLY.** At the end of section 2.1.2 we have added the following text: 'Uncertainties are assumed normally distributed, unless specifically indicated by IPCC-06. As discussed later, the distribution if transformed to log-normal after the aggregation'.

I have a difficulty understanding and interpreting Table 7. Could you give some additional explanation in the text what exactly the numbers in Table 7 mean and represent?

**REPLY.** We have improved the readability of table 7 (now table 6) and its description in the revised text.

Please use sufficiently large font size in all figures. I have difficulty reading them because they are so small. You may decide to leave out some labels to make more room for larger characters, and the figures easier on the eye.

**REPLY.** We have replotted all the figures to enhance readability.

The 'look' of the bar charts may be improved. The bars look very bulky and all very similar (e.g. Figure 2 etc.), only the colors vary. You might consider a different way of presenting the data, but this would be up to you.

**REPLY.** We have replotted all the figures and improved their readability.

Please make sure that the text explaining the figures and the captions (notably figures 13 and 14) is sufficient to easily understand what the figures show/represent and how to interpret them. It took me quite some time to 'sort of' understand them and this should not be necessary in my opinion. **REPLY.** Figure explanation and captions have been improved.